# Transformer Encoder Satisfiability: Complexity and Impact on Formal Reasoning

**Marco Sälzer**
University of Kaiserslautern-Landau, RPTU
Kaiserslautern, Germany
`marco.saelzer@rptu.de`

**Eric Alsmann**
Theoretical Computer Science / Formal Methods
University of Kassel, Germany
`eric.alsmann@uni-kassel.de`

**Martin Lange**
Theoretical Computer Science / Formal Methods
University of Kassel, Germany
`martin.lange@uni-kassel.de`

## Abstract

We analyse the complexity of the satisfiability problem, or similarly feasibility problem, (trSAT) for transformer encoders (TE), which naturally occurs in formal verification or interpretation, collectively referred to as formal reasoning. We find that trSAT is undecidable when considering TE as they are commonly studied in the expressiveness community. Furthermore, we identify practical scenarios where trSAT is decidable and establish corresponding complexity bounds. Beyond trivial cases, we find that quantized TE, those restricted by fixed-width arithmetic, lead to the decidability of trSAT due to their limited attention capabilities. However, the problem remains difficult, as we establish scenarios where trSAT is NEXPTIME-hard and others where it is solvable in NEXPTIME for quantized TE. To complement our complexity results, we place our findings and their implications in the broader context of formal reasoning.

## 1 Introduction

Natural language processing (NLP) models, processing and computing human language, are gateways for modern applications aiming to interact with human users in a natural way. Although NLP is a traditional field of research, the use of deep learning techniques has undoubtedly revolutionised the field in recent years (Otter et al. (2021)). In this revolution, models such as Recurrent Neural Networks (RNN) or more specific Long Short-term Memory Networks (LSTM) (Yu et al. (2019)) have long been the driving force, but for a few years now NLP has a new figurehead: *transformers* (Vaswani et al. (2017)).

Transformers are a deep learning model using (multiple) self-attention mechanisms to process sequential input data, usually natural language. The efficient trainability of transformers, for example in contrast to LSTM, while achieving top-tier performance led to numerous heavy-impact implementations such as BERT (Devlin et al. (2019)), GPT-3 (Brown et al. (2020)) or GPT-4 (OpenAI (2023)), sparking widespread use of the transformer architecture. However, the foreseeable omnipresence of transformer-based applications leads to serious security concerns.

In general, there are two approaches to establishing trustworthiness of learning-based models: first, certifying specific, application-dependent safety properties, called *verification*, and second, interpreting the behaviour of such models and giving explanations for it, called *interpretation*. In both approaches, the holy grail is to develop automatic methods that are *sound and complete*: algorithm $A$ given some model $T$ and (verification or interpretation) specification $\varphi$ outputs `true` if $T$ satisfies $\varphi$ (soundness) and for every given pair $T, \varphi$ where $T$ satisfies $\varphi$ algorithm $A$ outputs `true` (completeness). We refer to such sound and complete methods and tasks for verification and interpretation collectively using the term *formal reasoning*.

We lay out a framework for the possibilities and challenges of formal reasoning for transformers by establishing fundamental complexity (and computability) results in this work. Thereby, we focus on the so-called *satisfiability (*TRSAT*) problem* of sequence-classifying transformers: given a transformer $T$, decide whether there is some input word $w$ such that $T(w) = 1$, which can be interpreted as $T$ *accepts* $w$. Although this may seem like an artificial problem at first glance, it is a natural abstraction of problems that commonly occur in almost all non-trivial formal reasoning tasks. Additionally, since it is detached from the specifics of particular reasoning specifications like safety properties for instance, uncomputability results and complexity-theoretic hardness results immediately transfer to more complex formal reasoning tasks. This also keeps the focus on the transformer architecture under consideration. Here, we exclusively consider *transformer encoders* (TE), which are encoder-only. This is mainly due to the fact that the known high expressive power of encoder-decoder transformers (Pérez et al. (2021)) makes formal reasoning trivially impossible.

Our work is structured as follows. We define necessary preliminaries in Section 2. In Section 3, we give an overview on our complexity results and take a comprehensive look at their implications for formal reasoning for transformers. In Section 4 and Section 5 we present our theoretical results: we show that TRSAT is undecidable for classes of TE commonly considered in research on transformer expressiveness, we show that a bounded version BTRSAT of the satisfiability problem is decidable, for any class of (computable) TE, and give corresponding complexity bounds and we show that considering quantized TE, meaning TE whose parameters and computations are carried out in a fixed-width arithmetic, leads to decidability of TRSAT and give corresponding complexity bounds. Finally, we discuss limitations, open problems and future research in Section 6.

**Related work.** We establish basic computability and complexity results about transformer-related formal reasoning problems, like formal verification or interpretation. This places our work in the intersection between research on *verification and interpretation of transformers* and *transformer expressiveness*.

There is a limited amount of work concerned with methods for the verification of safety properties of transformers (Hsieh et al. (2019); Shi et al. (2020); Bonaert et al. (2021); Dong et al. (2021)). However, all those methods do not fall in the category of formal reasoning, as they are non-complete. This means, the rigorous computability and complexity bound established in this work cannot be applied without further considerations. The same applies for so far considered interpretability methods (Zhao et al. (2024)). We remark that a lot of these approaches are not sound methods either.

In contrast, there is an uprise in theoretical investigations of transformer expressiveness. Initial work dealt with encoder-decoder models and showed that such models are Turing-complete (Pérez et al. (2021); Bhattamishra et al. (2020)). Note that these are different models than the ones we consider, which are encoder-only. Encoder-only models have so far been analysed in connection with circuit complexity (Hahn (2020); Hao et al. (2022); Merrill et al. (2022); Merrill & Sabharwal (2023b)), logics (Chiang et al. (2023); Merrill & Sabharwal (2023a)) and programming languages (Weiss et al. (2021)). A recently published survey (Strobl et al. (2024)) provides an overview of these results. This work is adjacent as some of the here considered classes of TE, mainly those considered in Section 4, are motivated by these results and some of the constructions we use in corresponding proofs are similar.

## 2 FUNDAMENTALS

**Mathematical basics.** Let $\Sigma$ be a finite set of symbols, called *alphabet*. A *(finite) word $w$ over $\Sigma$* is a finite sequence $a_1 \cdots a_k$ where $a_i \in \Sigma$. We define $|w| = k$. As usual, we denote the set of all non-empty words by $\Sigma^+$. A *language* is a set of words. We also extend the notion of an alphabet to vectors $\boldsymbol{x}_i \in \mathbb{R}^d$, meaning that a sequence $\boldsymbol{x}_1 \cdots \boldsymbol{x}_k$ is a word over some subset of $\mathbb{R}^d$. Usually, we denote vectors using bold symbols like $\boldsymbol{x}, \boldsymbol{y}$ or $\boldsymbol{z}$.

**Transformer encoders (TE).** We consider the transformer encoders (TE), based on the transformer architecture originally introduced in (Vaswani et al. (2017)). We take a look at TE from a computability and complexity perspective, making a formal definition of the considered architecture necessary. Thereby, we follow the lines of works concerned with formal aspects of transformers like

(Hahn (2020); Pérez et al. (2021); Hao et al. (2022)). From a syntax point of view, our definition is most near to (Hao et al. (2022)).[1]

An *TE $T$ with $L$ layers and $h_i$ attention heads in layer $i$* is a tuple $(emb, \{att_{i,j} \mid 1 \leq i \leq L, 1 \leq j \leq h_i\}, \{comb_i \mid 1 \leq i \leq L\}, out)$ where

- $emb \colon \Sigma \times \mathbb{N} \to \mathbb{R}^{d_0}$ for some $d_0 \in \mathbb{N}$ is the *positional embedding*,
- each *attention head* is a tuple $att_{i,j} = (score_{i,j}, pool_{i,j})$ where $score_{i,j} \colon \mathbb{R}^{d_{i-1}} \times \mathbb{R}^{d_{i-1}} \to \mathbb{R}$ is a function called *scoring* and $pool_{i,j} \colon (\mathbb{R}^{d_{i-1}})^+ \times \mathbb{R}^+ \to \mathbb{R}^{d_i}$ is a function called *pooling*, computing $(\boldsymbol{x}_1, \ldots, \boldsymbol{x}_n, s_1, \ldots, s_n) \mapsto \sum_{i'=1}^{n} norm(i', s_1, \ldots, s_n)(W \boldsymbol{x}_{i'})$ where $W$ is a linear map represented by a matrix and $norm \colon \mathbb{N} \times \mathbb{R}^+ \to \mathbb{R}$ is a *normalisation*,
- each $comb_i \colon \mathbb{R}^{d_{i,1}} \times \cdots \times \mathbb{R}^{d_{i,h_1+1}} \to \mathbb{R}^{d_i}$ is called a *combination* and $out \colon \mathbb{R}^{d_L} \to \mathbb{R}$ is called the *output*.

For given $i \leq L$ we call the tuple $(att_{i,1}, \ldots, att_{i,h_i}, comb_i)$ the *$i$-th layer* of $T$. The TE $T$ computes a function $\Sigma^+ \to \mathbb{R}$ as follows, also schematically depicted in Figure 1. Let $w = a_1, \ldots, a_n \in \Sigma^+$ be a word. First, $T$ computes an embedding of $w$ by $emb(w) = \boldsymbol{x}_1^0 \cdots \boldsymbol{x}_n^0$ where $\boldsymbol{x}_i^0 = emb(a_i, i)$. Next, each layer $1 \leq i \leq L$ computes a sequence $\boldsymbol{x}_1^i \cdots \boldsymbol{x}_n^i$ as follows: for each input $\boldsymbol{x}_m^{i-1}$ and attention head $att_{i,j}$, layer $i$ computes $\boldsymbol{y}_{m,j}^i = pool_{i,j}(\boldsymbol{x}_1^{i-1}, \ldots, \boldsymbol{x}_n^{i-1}, score_{i,j}(\boldsymbol{x}_m^{i-1}, \boldsymbol{x}_1^{i-1}), \ldots, score_{i,j}(\boldsymbol{x}_m^{i-1}, \boldsymbol{x}_n^{i-1}))$. Then, $\boldsymbol{x}_m^i$ is given by $comb_i(\boldsymbol{x}_m^{i-1}, \boldsymbol{y}_{m,1}^i, \ldots, \boldsymbol{y}_{m,h_i}^i)$. In the end, the output $T(w)$ is computed by $out(\boldsymbol{x}_n^k)$, thus the value of the output function for the last symbol of $w$ after being transformed by the embedding and $L$ layers of $T$. We say that $T$ *accepts* $w$ if $T(w) = 1$, and we say that $T$ *rejects* $w$ otherwise. We call $L$ the *depth* of $T$ and the maximal $h_i$ the *(maximum) width* of $T$. Furthermore, we call the maximal $d_i$ the *(maximum) dimensionality* of $T$. Let $\mathcal{T}, \mathcal{T}'$ be some classes of TE. We sometimes say that $\mathcal{T}'$ *is at least as expressive as $\mathcal{T}$* or $\mathcal{T}$ *is at most as expressive as $\mathcal{T}'$*, meaning that for each $T \in \mathcal{T}$ there is $T' \in \mathcal{T}'$ such that $T$ and $T'$ compute the same function. The decision problem $\text{TRSAT}[\mathcal{T}]$ is given $T \in \mathcal{T}$ over alphabet $\Sigma$, decide whether there is $w \in \Sigma^+$ such that $T(w) = 1$. We refer to this as the *satisfiability problem* of $\mathcal{T}$.[2]

**Fixed-width arithmetics.** We consider commonly used *fixed-width arithmetics (FA)* that represent numbers using a fixed amount of bits, like floating- or fixed-point arithmetic in this work. See (Baranowski et al. (2020)) (fixed-point) or (Constantinides et al. (2021)) (floating-point) for rigorous mathematical definitions of such FA. In this work, however, we only make use of a high-level view on different FA. Namely, given some FA $F$ we assume that all values are represented in binary using $b \in \mathbb{N}$ bits for representing its numbers. Thus, there are $2^b$ different rational numbers representable in $F$. Furthermore, we assume that the considered FA can handle overflow situations using either saturation or wrap-around and rounding situations by rounding up or off. We consider TE in the context of $F$. We say that $T$ *works over $F$*, assuming that all computations as well as values occurring in a computation $T(w)$ are carried out in the arithmetic defined by $F$.

## 3 Overview of complexity results and connection to formal reasoning

We address elementary problems arising in formal reasoning for transformers in this work. In doing so, we pursue the goal of establishing basic computability and complexity results for corresponding problems in order to frame possibilities and challenges.

We want our results to be detached from any intricacies of specific transformer architectures: first, we focus on transformer encoders (TE), so leaving any decoder mechanism unconsidered. The primary reason for this is that encoder-decoder architectures are of such high expressive power

---

[1]As of now, no single definition of Transformer encoders (TE) has been universally adopted in research on their formal aspects, particularly concerning syntax (see the recent survey (Strobl et al. (2024)) for an overview of different notions of TE). The definition we use here is sufficiently general and provides a parameterized template for the classes of TE considered in our main results.

[2]We observe that we can equivalently define TRSAT as requiring $T(w) \geq c$ for arbitrary $c \in \mathbb{Q}$ without changing any of the results detailed in this work.

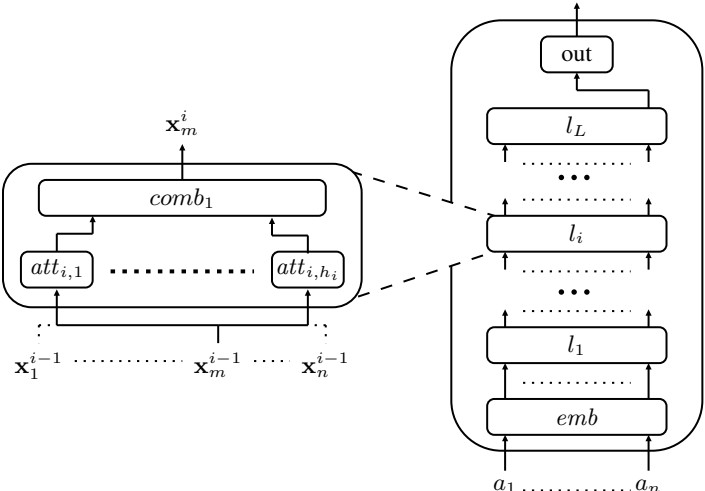

Figure 1: Schematic depiction of an TE $T$ with embedding $emb$ and $k$ (encoder) layers $l_i$. Each layer $l_i$ consists of some $h_i$ attention heads $att_{i,j}$, whose output is combined by $comb_i$. Additionally, for some layer $l_i$, the computational flow of $T$ regarding input position $m$ is schematically depicted in detail.

(Pérez et al. (2021)) that almost all formal reasoning problems are easily seen to be undecidable. The secondary reason for this is that encoder-decoder architectures subsume encoder-only architectures. So any lower complexity bound, established in this work, is also a lower bound for encoder-decoder transformers.

### SATISFIABILITY AS A BASELINE FORMAL RESONING PROBLEM

To achieve widespread implications of our results, we focus our considerations on a fundamental problem arising in formal verification and interpretation tasks: given a TE $T$, decide whether there is some input $w$ leading to some specific output $T(w)$, as defined formally in terms of the *satisfiability problem* TRSAT$[\mathcal{T}]$ for a class $\mathcal{T}$ of specific TE, see Section 2.

To see that this captures the essence of formal reasoning problems occurring in practice, consider the following formal verification task: Given a TE $T$, verify that $T$ only accepts inputs where every occurrence of a specific key from a set $K$ is accompanied by a particular pattern—for example, a key from $K$ must be immediately followed by a value from a set $V$. Such tasks are important to ensure syntactic correctness or adherence to some protocol specification. Formally, this is called a robustness property (Shi et al. (2020); Huang et al. (2023)). We can phrase this example task as a satisfiability problem by considering the property's negation, namely, to verify that there exists some input $w$ in which a key from $K$ is not properly followed by a value from $V$, yet we have $T(w) = 1$.

Similarly, consider a formal interpretation task where we aim to find the minimal subset $E' \subseteq E$ of some set of error symbols $E$ such that all inputs $w$ containing all errors in $E'$ are rejected by $T$. For instance, in a spam detection system powered by a transformer encoder, $E$ could represent a set of spam indicators or malicious keywords. We might want to determine the minimal combination of these indicators that will cause the system to classify an input as spam. This is understood as an abductive explanation in formal explainable AI (Marques-Silva & Ignatiev (2022)). Given a candidate subset $E'$, we can certify this by checking that there is some $w$ which contains all errors $E'$, but is accepted by $T$. This scenario is a special case of the satisfiability problem TRSAT$[\mathcal{T}]$ for some transformer class $\mathcal{T}$.

### OVERVIEW OF RESULTS ON THE COMPLEXITY OF TRANSFORMER ENCODER SATISFIABILITY

We start by considering the class $\mathcal{T}_{udec}$ of TE, motivated by commonly considered architectures in the theoretical expressiveness community (Pérez et al. (2021); Hao et al. (2022); Hahn (2020)): $\mathcal{T}_{udec}$ consists of those TE that use a positional embedding, expressive enough to compute a sum,

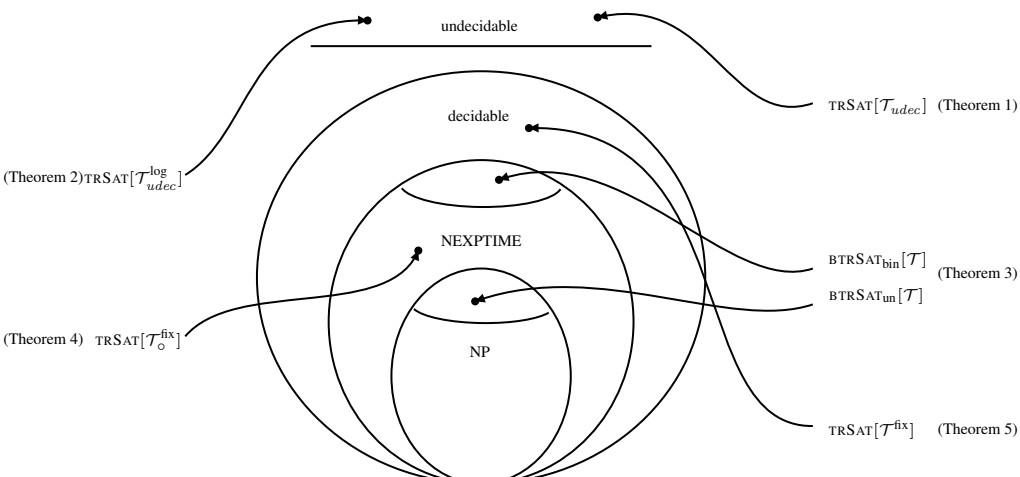

Figure 2: Schematic overview of the computability and complexity results, established in this work. The classes of TE are described in the pretext of the respective theorem. Note that $\mathcal{T}$ refers to an arbitrary class of (computable) TE. The small subset in the classes NP and NEXPTIME refers to the complete problems. The NEXPTIME-hardness result of $\text{TRSAT}[\mathcal{T}^{\text{FIX}}]$ is not visualized

hardmax $\texttt{hardmax}$ as normalisation functions and a scalar-product based scoring, enriched with a nonlinear map represented by an FNN.

**Theorem 1** (Section 4). *The satisfiability problem $\text{TRSAT}[\mathcal{T}_{udec}]$ is undecidable.*

Essentially, this result implies that even for TE the combination of $\texttt{hardmax}$ normalizations and expressive scoring is enough to make satisfiability undecidable. Generally, this makes formal reasoning, like verifying robustness properties or giving formal explanations, impossible for classes of TE that subsume $\mathcal{T}_{udec}$. Specifically, no such methods exist that are fully automatic, sound and complete. Theorem 1 does not preclude the existence of incomplete methods for instance.

Recently, so-called *log-precision transformers* have been studied (Merrill & Sabharwal (2023a)). These transformers are defined as usual, but given a word length $n$ it is assumed that a log-precision transformer $T$ uses at most $\mathcal{O}(\log(n))$ bits in its internal computations. To complement these theoretical considerations, we consider the class $\mathcal{T}_{udec}^{\text{LOG}}$ of TE from $\mathcal{T}_{udec}$ that work with log-precision. Unfortunately, this restriction is not enough to circumvent general undecidability.

**Theorem 2** (Section 4). *The satisfiability problem $\text{TRSAT}[\mathcal{T}_{udec}^{\text{LOG}}]$ is undecidable.*

Given such impossibility results, we turn our attention to the search for decidable cases. We make the reasonable assumption that all considered TE are computable, meaning that their components like scoring, normalisation, pooling, combination and output functions are computable functions. Moreover, we assume that each TE $T$ computes its output $T(w)$ for a given input $w$ within polynomial time relative to the size of $T$ and the length of $w$. This is reasonable, as the output is computed layer-wise where each layer involves a quadratic amount of calculations per attention head. Therefore, the computation depends polynomially on the depth and width of $T$ and the length of $w$.

First, we consider a natural restriction of the satisfiability problem by bounding the length of valid inputs. Then satisfiability becomes decidable, regardless of the respective class of TE, but it is difficult from a complexity-theoretic perspective. To formalize this, we introduce the *bounded satisfiability problem* $\text{BTRSAT}[\mathcal{T}]$ for a class $\mathcal{T}$: given an TE $T \in \mathcal{T}$ and a bound $n \in \mathbb{N}$ on its input length, decide whether there is word $w$ with $|w| \leq n$ s.t. $T(w) = 1$.

**Theorem 3** (Section 5, informal restatement). *The bounded satisfiability problem $\text{BTRSAT}[\mathcal{T}]$ is decidable for all classes $\mathcal{T}$ of (computable) TE. Depending on whether $n$ is given in binary or unary coding, $\text{BTRSAT}[\mathcal{T}]$ is NEXPTIME-, resp. NP-complete assuming $\mathcal{T} \supseteq \mathcal{T}_{udec}$.*

Informally, this result implies that bounding the word length is a method to enable formal reasoning. However, it does not change the fact that satisfiability is an essentially hard problem. As hardness is a lower bound, this also translates to subsuming formal reasoning tasks.

Imposing a bound on the input length may not be a viable restriction for various formal reasoning tasks. We therefore study other ways of obtaining decidability. We address the unbounded satisfiability problem for practically motivated classes of TE. We consider the class $\mathcal{T}_\circ^{\text{FIX}}$ of TE that use a positional embedding with some periodicity in their positional encoding, commonly seen in practice (Vaswani et al. (2017); Dufter et al. (2022)), use softmax or hardmax as normalisation and which work over some fixed-width arithmetic (FA). This last restriction is motivated by recent popular ways to handle ever increasing TE sizes, for example via quantization or using low-bit arithmetics (Bondarenko et al. (2021)). From a complexity-theoretic perspective, the use of fixed-width arithmetic has a similar effect to bounding the input length.

**Theorem 4** (Section 5). *The satisfiability problem* $\text{TRSAT}[\mathcal{T}_\circ^{\text{FIX}}]$ *is in NEXPTIME.*

So automatic, sound and complete formal reasoning for periodical TE in a fixed-width arithmetic environment is generally possible with potentially high complexity. Note that formal reasoning tasks with more complex safety or interpretability specifications than simple satisfiability may even lead to higher complexities.

We then aim to show that this is optimal by providing a matching lower bound. However, we need to relax these restrictions again, namely considering the class $\mathcal{T}^{\text{FIX}}$ allowing for TE that use arbitrary embeddings and work over some fixed-width arithmetic. However, due to the fixed-witdth arithmetic assumption, which consequently applies to positional informations as well, every embedding must necessarily witness a periodic behaviour. Thus, decidability is implied by the same arguments as used in Theorem 4. Additionally, we show that high complexity is unavoidable, making sound and complete automatic formal reasoning for fixed-width arithmetic transformers with arbitrary positional embeddings practically intractable.

**Theorem 5** (Section 5). *The satisfiability problem* $\text{TRSAT}[\mathcal{T}^{\text{FIX}}]$ *is decidable and NEXPTIME-hard.*

Figure 2 provides a schematic illustration of the computability and complexity results summarized in this section. This figure is intended purely for technical clarity, summarizing our findings without delving into the formal reasoning implications discussed earlier.

## 4 TRANSFORMER ENCODER SATISFIABILITY IS GENERALLY UNDECIDABLE

We consider a class of TE, denoted by $\mathcal{T}_{udec}$, which we design with the aim of minimising its expressive power, but having an undecidable satisfiability problem. We define $\mathcal{T}_{udec}$ by giving minimum requirements: positional-embeddings can be of the form $emb(a_k, 0) = (1, 1, 0, 0, k)$ and $emb(a_k, i) = (0, 1, i, \sum_{j=0}^{i} j, k)$ where we assume some order on the alphabet symbols $a_1, a_2, \ldots$. For scoring functions we allow for $N(\langle Q\boldsymbol{x}, K\boldsymbol{y} \rangle)$ where $N$ is a classical Feedforward Neural Network (FNN) with $relu$ activations, $Q$ and $K$ are linear maps and $\langle \cdots \rangle$ denotes the usual scalar product, for normalisations we allow for hardmax $\text{hardmax}(i, x_1, \ldots, x_n) = \frac{1}{m}$ if $x_i \geq x_j$ for all $j \leq n$ and there are $m$ distinct $x_j$ such that $x_i = x_j$ otherwise $\text{hardmax}(i, x_1, \ldots, x_n) = 0$. Combinations as well as output functions can be classical FNN with $relu$ activation. Aside from technical reasons, we motivate the choice of $\mathcal{T}_{udec}$ in Section 3. To ease our notation, we exploit the fact that using hardmax as normalisation implies a clearly defined subset of positions $M$ that are effective in the computation of some attention head $att$ given some position $i$, namely those that are weighted non-zero. In this case, we say that $att$ *attends to* $M$ *given position* $i$.

We prove that $\text{TRSAT}[\mathcal{T}_{udec}]$ is undecidable by establishing a reduction from the *(unbounded) octant tiling-word problem (*OTWP*[*])*. For details on tiling problems, see Appendix A. The OTWP[*] is defined as follows: given a tiling system $\mathcal{S} = (S, H, V, t_I, t_F)$ where $S$ is some finite set of tiles, $H, V \subseteq S^2$ and $t_I, t_F \in S$ we have to decide whether there is a word (a) $t_{0,0}, t_{1,0}, t_{1,1}, t_{2,0}, t_{2,1}, t_{2,2}, t_{3,0}, \ldots, t_{k,k} \in S^+$ such that (b) $t_{0,0} = t_I, t_{k,k} = t_F$, (c) for all $i \leq k$ and $0 \leq j < i$ holds $(t_{i,j}, t_{i,j+1}) \in H$ and (d) for all $i \leq k-1$ and $j \leq i$ holds $(t_{i,j}, t_{i+1,j}) \in V$. We call a word $w$ which satisfies (a) an *encoded tiling* and if (b)-(d) are satisfied as well then we call $w$ a *valid* encoded tiling. Our proof strategy is easily described: given a tiling system $\mathcal{S}$, we build an TE $T_\mathcal{S} \in \mathcal{T}_{udec}$ which accepts a word $w$ if it fulfils conditions (a) to (d) and otherwise $T_\mathcal{S}$ rejects $w$.

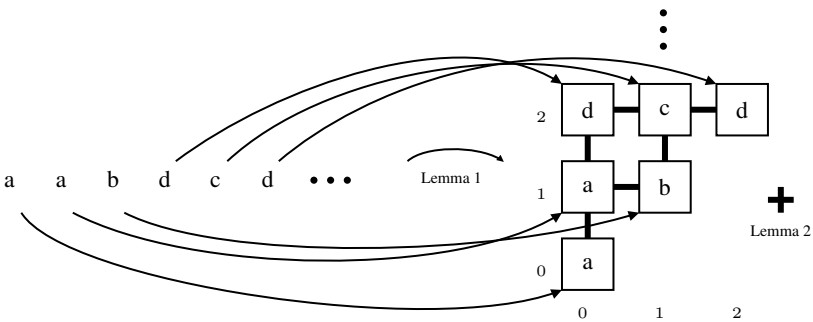

Figure 3: Schematic depiction of the expressive capabilities of TE from $\mathcal{T}_{udec}$ in context of the OTWP* proven in Lemma 1 and Lemma 2.

We derive most technical proofs of the following lemmas and theorems to Appendix B and instead provide intuitions and proof sketches in this section.

We start with the first observation: the expressiveness of TE in $\mathcal{T}_{udec}$ is sufficient to decode the octant tiling potentially represented by a given word $w$, as depicted by the arrows in Figure 3. In detail, two encoder layers in combination with a positional embedding definable in $\mathcal{T}_{udec}$ are expressive enough to compute for a given symbol $t$ in $w$ to which position in an octant tiling it corresponds, if we interpret $w$ as an encoded tiling.

**Lemma 1.** *Let $\mathcal{S}$ be a tiling system with tiles $S = \{a_1, \ldots, a_k\}$. There is an embedding function $emb$ and there are encoder layers $l_1$ and $l_2$ definable in $\mathcal{T}_{udec}$ such that for each word $w = t_{0,0}t_{1,0}t_{1,1}t_{2,0}\cdots t_{m,n} \in S^+$ holds that $l_2(l_1(emb(w))) = \boldsymbol{x}_1^2 \ldots \boldsymbol{x}_{|w|}^2$ where $\boldsymbol{x}_i^2 = (1, i, r(i), c(i), k_i)$ such that $a_{k_i}$ is equal to the symbol at position $i$ in $w$ and $(r(1), c(1)), (r(2), c(2)), \ldots, (r(|w|), c(|w|))$ is equal to $(0,0), (1,0) \ldots, (m,n)$.*

Assume that $w \in S^+$. Lemma 1 implies that a TE $T \in \mathcal{T}_{udec}$ is generally able to recognize whether $w$ is an encoded tiling as soon as $T$ is able to check whether $r(|w|)$ and $c(|w|)$ of the last symbol of $w$ processed by $l_2(l_1(emb(\cdots)))$ are equal. Therefore, property (a) and also (b) can be checked by TE in $\mathcal{T}_{udec}$ using the residual connection and FNN.

Property (c) can be ensured if it is possible to build an attention head that is able to attend to position $k+1$ given position $k$. Let $w = t_{0,0}t_{1,0}t_{1,1}t_{2,0}\cdots t_{m,m}$ with $t_{i,j} \in S$. To verify whether property (d) holds, a TE must be able to attend to position $k + (i+1)$ given position $k$ corresponding to symbol $t_{i,j}$. This is depicted in Figure 3 by the bold lines between horizontal and vertical tiles. In summary, to check properties (a) – (d) it is left to argue that there are attention heads in $\mathcal{T}_{udec}$ that can attend to positions depending linearly on the values of the currently considered position.

**Lemma 2.** *Let $f(x_1, \ldots, x_k) = a_1 x_1 + \cdots + a_k x_k + b$ with $a_i, b \in \mathbb{R}$ be some linear function. There is attention head $att_f$ in $\mathcal{T}_{udec}$ such that for all sequences $\boldsymbol{x}_1, \ldots, \boldsymbol{x}_m$ where all $\boldsymbol{x}_i = (1, i, \boldsymbol{y}_i)$ for some $\boldsymbol{y}_i \in \mathbb{R}^{k-2}$ attention head $att_f$ attends to $\{\boldsymbol{x}_j, \boldsymbol{x}_{j+1}\}$ given position $i$ if $f(\boldsymbol{x}_i) = j + \frac{1}{2}$ with $j \leq m-1$ and otherwise to $\{\boldsymbol{x}_j\}$ where $j$ is the value nearest to $f(\boldsymbol{x}_i)$.*

In combination, the previous lemmas indicate that TE from $\mathcal{T}_{udec}$ are able to verify whether a given word is a valid encoded tiling. This expressive power is enough, to lead to an undecidable satisfiability problem for TE from $\mathcal{T}_{udec}$.

**Theorem 1.** *The decision problem $\text{TRSAT}[\mathcal{T}_{udec}]$ is undecidable.*

*Proof Sketch.* We establish a reduction from OTWP* to $\text{TRSAT}[\mathcal{T}_{udec}]$ by constructing for each instance $\mathcal{S} = (S, H, V, t_I, t_F)$ of OTWP* an TE $T_\mathcal{S}$ accepting exactly those $w$ corresponding to a valid encoded-tiling for $\mathcal{S}$.

$T_\mathcal{S}$ uses the positional embedding described in the beginning of Section 4 and has four layers. Layers $l_1$ and $l_2$ are given by Lemma 1 and are used to decode the row and column indexes corresponding to a potential octant tiling for each symbol in a given word $w$. Layer $l_3$ uses the informations encoded by the embedding and the decoded row and column indexes to check whether properties (a) to (d)

described above hold for $w$. The necessary informations are aggregated using three attention heads $att_{prev}$, $att_{next}$ and $att_{step}$, each built according to Lemma 2. Thereby, $att_{prev}$ attends each position to its predecessor, but the first position attends to itself. This allows to clearly identify the vector corresponding to the first position in $w$ and check whether this is equal to tile $t_I$. Attention head $att_{next}$ attends each position to its successor, but the last position attends to itself. This allows to clearly identify the vector corresponding to the last position in $w$, in order to check whether this is equal to $t_F$, and to check conditions given by $H$. Attention head $att_{step}$ attends each position to the position with the same column index but the successive row index. If there is no such successive row it attends to the last position. This allows to check whether conditions given by $V$ holds. Each of these conditions is checked in the combination function of $l_3$, using specifically built feed-forward neural networks outputting 0 to some predefined vector dimension if and only if the condition is met. Finally, layer $l_4$ aggregates the information of all positions in the vector corresponding to the last position using attention head $att_{leq}$, again given by Lemma 2. The correctness of this reduction follows from the detailed construction of $T_S$, given in Appendix B. $\qquad\square$

Next, we consider the class $\mathcal{T}_{udec}^{\text{LOG}}$ which is defined exactly like $\mathcal{T}_{udec}$ but for all $T \in \mathcal{T}_{udec}^{\text{LOG}}$ working over alphabet $\Sigma$ and all words $w$ with $|w| = n$ we assume that $T(w)$ is carried out in some fixed-width arithmetic $F$ using $\mathcal{O}(\log(\max(|\Sigma|, n)))$ bits.

**Theorem 2.** *The decision problem* $\text{TRSAT}[\mathcal{T}_{udec}^{\text{LOG}}]$ *is undecidable.*

*Proof sketch.* This proof follows the exact same line as the proof of Theorem 1. Additionally, we need to argue that $T_S$ works as intended, despite the fact that it is limited by some log-precision $F$.

Looking at the proof of Theorem 1, it is imminent that the magnitude and precision of all values used and produced in the computation $T_S(w)$ depend polynomially on $n$ and, thus, we can choose the representation of $F$ to be linear in $\log(n)$, which avoids any overflow or rounding situations and ensures that $T_S$ works as intended. A formal proof is given in Appendix B. $\qquad\square$

## 5    HOW TO MAKE TRANSFORMER ENCODER SATISFIABILITY DECIDABLE

In this section we investigate classes of TE leading to decidable TRSAT problems or decidable restrictions of it. Additionally, we establish corresponding complexity bounds.

In order to establish clearly delineated upper complexity bounds, we need to bound the representation size of a TE $T$. Instead of tediously analyzing the space needed to represent embedding, scoring, pooling, combination and normalisation functions, we note that it suffices to estimate the size up to polynomials only. The *complexity* of a TE $T$ with $L$ layers and $h_i$ attention heads in layer $i$, working on inputs over alphabet $\Sigma$, is $|T| := |\Sigma| + L + H + D$ where $H := \max\{h_i \mid 1 \le i \le L\}$ and $D$ is the maximal dimensionality of vectors occurring in a computation of $T$. Note that one can reasonably assume the *size* of a syntactic representation of $T$ to be polynomial in $|T|$, and that TE have the *polynomial evaluation property*: given a word $w \in \Sigma^+$, $T(w)$ can be computed in time that is polynomial in $|T| + |w|$. Section 3 discusses why this assumption is reasonable.

We start with a natural restriction: bounding the word length. Let $\mathcal{T}$ be a class of TE. The *bounded satisfiability problem*, denoted by $\text{BTRSAT}[\mathcal{T}]$ is: given $T \in \mathcal{T}$ and some $n \in \mathbb{N}$, decide whether there is a word $w$ with $|w| \le n$ such that $T(w) = 1$. It is not hard to see that $\text{BTRSAT}[\mathcal{T}]$ is decidable. However, its complexity depends on the value of $n$, and we therefore distinguish whether $n$ is represented in *binary* or *unary* encoding. We denote the problems as $\text{BTRSAT}_{\text{bin}}[\mathcal{T}]$ and $\text{BTRSAT}_{\text{un}}[\mathcal{T}]$.

**Theorem 3.** *Let $\mathcal{T}$ be a class of TE. Then*

1. $\text{BTRSAT}_{\text{un}}[\mathcal{T}]$ *is in NP; if $\mathcal{T}_{udec} \subseteq \mathcal{T}$ then $\text{BTRSAT}_{\text{un}}[\mathcal{T}]$ is NP-complete,*

2. $\text{BTRSAT}_{\text{bin}}[\mathcal{T}]$ *is in NEXPTIME; if $\mathcal{T}_{udec} \subseteq \mathcal{T}$ then $\text{BTRSAT}_{\text{bin}}[\mathcal{T}]$ is NEXPTIME-complete.*

*Proof Sketch.* The decidability result of statement (1) can be shown using a simple guess-and-check argument: given $n \in \mathbb{N}$, guess a word $w \in \Sigma^+$ with $|w| \le n$, compute $T(w)$ and check that the result is 1. This is possible in time polynomial in $|T| + n$ using the polynomial evaluation property. Note that $|T|$ depends on $|\Sigma|$, thus this also respects the actual representation size of $w$.

Moreover, the value of $|T| + n$ is polynomial in the size needed to represent $n$ in unary encoding. The decidability result of statement (2) is shown along the same lines. However, if the value $n$ is encoded binarily then this part of the input is of size $\log n$, and $|T| + n$ becomes exponential in this. Hence, the guess-and-check procedure only proves that $\text{BTRSAT}_{\text{bin}}[\mathcal{T}] \in \text{NEXPTIME}$.

For the completeness result in (1) it suffices to argue that the problem is NP-hard. We use that TE in $\mathcal{T}_{udec}$ are expressive enough to accept a given word $w$ if and only if it is a valid encoded tiling, cf. Section 4 for details. It is possible to establish NP-hardness of a restriction of the octant word-tiling problem, namely the *bounded octant word-tiling problem* (for unarily encoded input values). See Appendix A for details on tiling problems. It remains to observe that the construction in Theorem 1 is in fact a polynomial-time reduction, and that it reduces the bounded octant word-tiling problem to the bounded satisfiability problem. The argument for NEXPTIME-hardness in statement (2) works the same with, again, the bounded octant-word tiling problem shown to be NEXPTIME-hard when the input parameter $n$ is encoded binary. A full proof for Theorem 3 is in Appendix C. □

We turn our attention to classes of TE that naturally arise in practical contexts. We consider TE that work over some fixed-width arithmetic, like fixed- or floating-point numbers, and which have an embedding relying on a periodical encoding of positions. We start with establishing a scenario where TRSAT is decidable in NEXPTIME. Regardless of the underlying TE class $\mathcal{T}$, our proof strategy always relies on a certifier-based understanding of NEXPTIME: given $T \in \mathcal{T}$, we nondeterministically guess a word $w$, followed by a deterministic certification whether $T(w) = 1$ holds. For this to show $\text{TRSAT}[\mathcal{T}] \in \text{NEXPTIME}$, we need to argue that the overall running time of such a procedure is at most exponential, in particular that whenever there is a word $w$ with $T(w) = 1$ then there is also some $w'$ with $T(w') = 1$ and $|w'| \leq 2^{poly(|T|)}$. Again, we rely on the polynomial evaluation property of TE in $\mathcal{T}$, i.e. the fact that $T(w')$ can be computed in time polynomial in $|T| + |w'|$.

We consider the class of TE $\mathcal{T}_\circ^{\text{FIX}}$, defined by placing restrictions on the positional embedding of an TE $T$ to be *additive-periodical* which means that $emb(a, i) = emb'(a) + pos(i)$ where $pos$ is periodical, i.e. there is $p \geq 1$ such that $pos(i) = pos(i + p)$ for all $i \in \mathbb{N}$. Additionally, all normalisation functions are realised by either the softmax function $\text{softmax}$ or the hardmax function $\text{hardmax}$. Moreover, we assume that all computations occurring in $T$ are carried out in some fixed-width arithmetic, encoding values in binary using a fixed number $b \in \mathbb{N}$ of bits. Aside from technical reasons, we motivate the choice of $\mathcal{T}_\circ^{\text{FIX}}$ in Section 3. Given these restrictions, we adjust the definition of the complexity of $T \in \mathcal{T}_\circ^{\text{FIX}}$ as a measure of the size (up to polynomials) as $|T| := |\Sigma| + L + H + D + p + b$.

**Lemma 3.** *There is a polynomial function* $poly \colon \mathbb{N} \to \mathbb{N}$ *such that for all* $T \in \mathcal{T}_\circ^{\text{FIX}}$ *and all words* $w$ *with* $T(w) = 1$ *there is word* $w'$ *with* $T(w') = 1$ *and* $|w'| \leq 2^{poly(|T|)}$.

*Proof Sketch.* The polynomial $poly$ can be chosen uniformly for all $T \in \mathcal{T}_\circ^{\text{FIX}}$ because for all positional embeddings of TE in $\mathcal{T}_\circ^{\text{FIX}}$ there is an upper bound on the period and on the bit-width in the underlying arithmetic. The small-word property stated by the lemma is then shown by arguing, given polynomial $poly$, TE $T$ and $|w| > 2^{poly(|T|)}$, that $w$ contains unnecessary subwords $u$ that can be cut out without changing the output in $T$. Here, we exploit the fact $T$ has some periodicity $p$ and only consider those $u$ whose length is a multitude of $p$. This ensures that the resulting word $w'$, given by $w$ without $u$, is embedded the same way as $w$ by the positional embedding of $T$. The existence of such subwords follows from $T$'s limited distinguishing capabilities, especially in its normalisations, due to the bounded representation size of numerical values possible in the underlying fixed-width arithmetic. A formal proof relies on basic combinatorial arguments and given in Appendix C. □

**Theorem 4.** $\text{TRSAT}[\mathcal{T}_\circ^{\text{FIX}}]$ *for additive-periodical TE over fixed-width arithmetic is in NEXPTIME.*

*Proof.* Let $T \in \mathcal{T}_\circ^{\text{FIX}}$ working over alphabet $\Sigma$. We use a certifier-based understanding of a nondeterministic exponential-time algorithm as follows: We (a) guess an input $w \in \Sigma^+$ and (b) compute $T(w)$ to check whether $T(w) = 1$. For correctness, we need to argue that the length of $w$ is at most exponential in $|T|$. This argument is given by Lemma 3. Note that via assumption we have that $T(w)$ can be computed in polynomial time regarding $|T|$ and $|w|$. □

Next, we address the goal of obtaining a matching lower bound, i.e. NEXPTIME-hardness. An obvious way to do so would be to follow Theorem 3.2 and form a reduction from the bounded

octant word-tiling problem. Hence, given a tiling system $\mathcal{S}$ and $n \in \mathbb{N}$ encoded binarily, we would have to construct – in time polynomial in $|\mathcal{S}| + \log n$ – an TE $T_{\mathcal{S},n} \in \mathcal{T}_\circ^{\text{FIX}}$ such that $T_{\mathcal{S},n}(w) = 1$ for some $w \in \Sigma^+$ iff there is a word $w = t_{1,1}, t_{2,1}, t_{2,2}, t_{3,1}, \ldots, t_{n,n}$ representing a valid $\mathcal{S}$-tiling. In particular, $T_{\mathcal{S},n}$ would have to be able to recognise the correct word length and reject input that is longer than $|w| = \frac{n(n+1)}{2}$. This poses a problem for TE with periodical embeddings. To recognize whether a word is too long, an TE $T$ must ultimately rely on its positional embedding, which seems to make a periodicity of $p \geq \frac{n(n+1)}{2}$ necessary. Since the size of periodical TE is linear in $p$, we get an exponential blow-up in a potential reduction of $\text{OTWP}_{\text{bin}}$ to $\text{TRSAT}[\mathcal{T}_\circ^{\text{FIX}}]$, given that the values of $\frac{n(n+1)}{2}$ and already $n$ are exponential in the size of a binary representation of $n$. This problem vanishes when the requirement of the underyling positional embedding to be periodical is lifted: allowing for arbitrary TE, working over some fixed-width arithmetic, leads to an NEXPTIME-hard satisfiability problem. Let $\mathcal{T}^{\text{FIX}}$ be defined similar to $\mathcal{T}_\circ^{\text{FIX}}$, but we allow for arbitrary embeddings. Furthermore, we assume that the considered fixed-width arithmetics can handle overflow situations using saturation.

**Theorem 5.** $\text{TRSAT}[\mathcal{T}^{\text{FIX}}]$ *for TE over fixed-width arithmetic is decidable and NEXPTIME-hard.*

*Proof sketch.* The decidability follows from the same arguments as in Theorem 4, with the insight that even though $T \in \mathcal{T}^{\text{FIX}}$ uses an arbitrary embedding, a fixed-width setting with $b$ bits enforces a periodicity of size at most $2^b$, assuming overflow is handled by wrap-around, or a periodicity of size 1 after a finite prefix of length up to $2^b$, assuming overflow is handled by saturation, in the embedding. For the hardness, we establish a reduction from $\text{OTP}_{\text{bin}}$ to $\text{TRSAT}[\mathcal{T}^{\text{FIX}}]$ by constructing, for each instance $(\mathcal{S}, n)$ of $\text{OTP}_{\text{bin}}$, an TE $T_{\mathcal{S},n}$ working over some fixed-width arithmetic $F$, which accepts exactly those $w$ with $|w| = \frac{n(n+1)}{2}$ corresponding to a valid word-encoded tiling for $\mathcal{S}$. See Appendix A for details on tiling problems. The construction is similar to the one given for $T_{\mathcal{S}}$ in the proof of Theorem 4, but we need to enable $T_{\mathcal{S},n}$ to reject words that are too long corresponding a polynomial bound dependent on $n$. This implies that $T_{\mathcal{S},n}$, based on the positional embedding $emb$ specified in Section 4, is able to check for all symbols if their respective position is less than or equal to a predefined bound. This can be achieved with similar tools as used in Lemma 2. Furthermore, we need to ensure that $T_{\mathcal{S},n}$ works as intended, despite the fact that it is limited by $F$. The arguments follow the same line as the proof of Theorem 2. A formal proof is given in Appendix C. $\qquad\square$

## 6    SUMMARY, LIMITATIONS AND OUTLOOK

We investigated the satisfiability problem of transformer encoders (TE). In particular, we considered the computational complexity of the satisfiability problem TRSAT of TE in context of different classes of TE, forming a baseline for understanding possibilities and challenges of formal reasoning of transformers. We showed that TRSAT is undecidable for classes of TE recently considered in research on the expressiveness of different transformer models (Theorem 1 and Theorem 2). This implies that formal reasoning is impossible as soon as we consider classes of TE that are at least as expressive as the classes considered in these results. Additionally, we identified two ways to enable formal reasoning for TE: by bounding the length of inputs (Theorem 3) or by considering quantized TE, where computations and parameters are limited by fixed-width arithmetic (Theorem 4). These imply that formal reasoning is possible for TE classes that are at most as expressive as those in our results. Thereby, we assume that TE expressiveness is the primary factor influencing computability or complexity bounds, rather than specific safety or interpretability assumptions. However, in both cases, TRSAT remains computationally difficult (Theorems 3 and 5). Again, these results apply only to TE classes at least as expressive as those we considered. While our results provide an initial framework for understanding the possibilities and challenges of formal reasoning for transformers, there is room for more detailed investigations. Our undecidability and hardness results rely on normalizations realized by the hardmax function, and it's unclear whether similar results hold when using the commonly employed softmax function. Additionally, further exploration of the interplay between the embedding function and the internal structure of the TE is of interest. We expect that less expressive embeddings require a richer attention mechanism, but it's unclear where the limits lie regarding the undecidability of the satisfiability problem. Regarding our decidability and upper complexity bounds, examining the specifics of particular fixed-width arithmetics could be practically beneficial. While this wouldn't change our overall results, it could provide tighter time-complexity estimates valuable for certain formal reasoning applications.

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

## A  TILING PROBLEMS

We make use of particular tiling problems in order to prove lower bounds on the complexity and decidability of $\text{TRSAT}[\mathcal{T}]$ for different classes $\mathcal{T}$.

A *tiling system* is an $\mathcal{S} = (S, H, V, t_I, t_F)$ where $S$ is a finite set; its elements are called *tiles*. $H, V \subseteq S \times S$ define a horizontal, resp. vertical matching relation between tiles, and $t_I, t_F$ are two designated *initial*, resp. *final* tiles in $S$.

Problems associated with tiling systems are typically of the following form: given a discrete convex plain consisting of cells with horizontal and vertical neighbors, is it possible to cover the plane with tiles from $S$ in a way that horizontally adjacent tiles respect the relation $H$ and vertically adjacent tiles respect the relation $V$, together with some additional constraints about where to put the initial and final tile $t_I, t_F$. Such tiling problems, in particular for rectangular planes, have proved to be extremely useful in computational complexity, cf. (Berger (1966); van Emde Boas (1997)), since they can be seen as abstract versions of halting problems.

We need a variant in which the plane to be tiled is of triangular shape. The *n-th triangle* is $\mathcal{O}_n = \{(i, j) \in \mathbb{N} \times \mathbb{N} \mid j \leq i \leq n\}$ for $n > 0$. An (*$\mathcal{S}$*)-tiling of $\mathcal{O}_n$ is a function $\tau : \mathcal{O}_n \to S$ s.t.

- $(\tau(i, j), \tau(i, j + 1)) \in H$ for all $(i, j) \in \mathcal{O}$ with $j < i \leq n$,
- $(\tau(i, j), \tau(i + 1, j)) \in V$ for all $(i, j) \in \mathcal{O}$ with $j \leq i < n$.

Such a tiling a *successful*, if additionally $\tau(0, 0) = t_I$ and $\tau(i, i) = t_F$ for some $(i, i) \in \mathcal{O}_n$.

The *unbounded octant tiling problem* (OTP*) is: given a tiling system $\mathcal{S}$, decide whether a successful $\mathcal{S}$-tiling of $\mathcal{O}_n$ exists for some $n \in \mathbb{N}$. The *bounded octant tiling problem* (OTP) is: given a

tiling system $\mathcal{S}$ and an $n \geq 1$, decide whether a successful $\mathcal{S}$-tiling of $\mathcal{O}_n$ exists. Note that here, $n$ is part of the input, and that it can be represented differently, for example in binary or in unary encoding. We distinguish these two cases by referring to OTP$_{\mathsf{bin}}$ and OTP$_{\mathsf{un}}$.

It is well-known that OTP* is undecidable (van Emde Boas (1997)). It is also not hard to imagine that OTP$_{\mathsf{un}}$ is NP-complete while OTP$_{\mathsf{un}}$ is NEXPTIME-complete. In fact, this is well-known for the variants in which the underlying plane is not a triangle of height $n$ but a square of height $n$ (van Emde Boas (1997)). The exponential difference incurred by the more compact binary representation of the input parameter $n$ is best seen when regarding the upper complexity bound for these problems: given $n$, a nondeterministic algorithm can simply guess all the $n^2$ many tiles of the underlying square and verify the horizontal and vertical matchings in time $\mathcal{O}(n^2)$. If $n$ is encoded unarily, i.e. the space needed to write it down is $s := n$, then the time needed for this is polynomial in the input size $s$; if $n$ is encoded binarily with space $s := \lceil \log n \rceil$ then the time needed for this is exponential in $s$.

It then remains to argue that the tiling problems based on triangular planes are also NP- resp. NEXPTIME-complete. Clearly, the upper bounds can be established with the same guess-and-check procedure. For the lower bounds it suffices to observe that hardness of the tiling problems for the squares is established by a reduction from the halting problem for Turing machines (TM) such that a square of size $n \times n$ represents a run of the TM of length $n$ as a sequence of rows, and each row represents a configuration of the TM using at most $n$ tape cells. This makes use of the observation that the space consumption of a TM can never exceed the time consumption. Likewise, assuming that a TM always starts a computation with its head on the very left end of a tape, one can easily observe that after $i$ time steps, it can change at most the $i$ leftmost tape cells. Hence, a run of a TM can therefore also be represented as a triangle with its first configuration of length 1 in row 1, the second of length 2 in row 2 etc.

At last, we consider two slight modifications of these two problems which are easily seen to preserve undecidability resp. NP- and NEXPTIME-completeness. The *unbounded octant tiling-word problem* (OTWP*) is: given some $\mathcal{S} = (S, H, V, t_I, t_F)$, decide whether there is a word $t_{0,0}, t_{1,0}, t_{1,1}, t_{2,0}, t_{2,1}, t_{2,2}, \ldots, t_{n,n} \in S^*$ for some $n \in \mathbb{N}$, s.t. the tiling $\tau$ defined by $\tau(i, j) := t_{i,j}$ comprises a successful tiling of $\mathcal{O}_n$. The two variants of the *bounded octant tiling-word problem* are both: given some $\mathcal{S}$ as above and $n$, decide whether such a word exists. Note that, again, here $n$ is an input parameter, and so its representation may affect the complexity of the problem, leading to the distinction between OTWP$_{\mathsf{bin}}$ with binary encoding and OTWP$_{\mathsf{un}}$ with unary encoding.

**Theorem 6.**

    *a)* OTWP* *is undecidable ($\Sigma_0^1$-complete).*

    *b)* OTWP$_{\mathsf{bin}}$ *is NEXPTIME-complete.*

    *c)* OTWP$_{\mathsf{un}}$ *is NP-complete.*

*Proof.* (a) It should be clear that a tiling problem and its tiling-word variant (like OTP* and OTWP*) are interreducible since they only differ in the formulation of how the witness for a successful tiling should be presented. So they are essentially the same problems. Undecidability of OTP* and, thus, OTWP* is known from (van Emde Boas (1997)), the $\Sigma_0^1$-upper bound can be obtained through a semi-decision procedure that searches through the infinite space of $\mathcal{O}_n$-tiling for any $n > 1$. This justifies the statement in part (a) of Thm. 6.

(b) With the same argument as in (a) t suffices to consider OTP$_{\mathsf{bin}}$ instead of OTWP$_{\mathsf{bin}}$. The upper bound is easy to see: a nondeterministic procedure can easily guess a tiling for $\mathcal{O}_n$ and verify the horizontal and vertical matching conditions, as well as the use of the initial and final tile in appropriate places. This is possible in time $\mathcal{O}(n^2)$, resp. $\mathcal{O}(2^{2\log n})$ which is therefore exponential in the input size $\lceil \log n \rceil$ for binarily encoded parameters $n$. This shows inclusion in NEXPTIME.

For the lower bound we argue that the halting problem for nondeterministic, exponentially-time bounded TM can be reduced to OTP$_{\mathsf{bin}}$: given a nondeterministic TM $\mathcal{M}$ over input alphabet $\Sigma$ and tape alphabet $\Gamma$ that halts after at most time $2^{p(n)}$ steps on input words of length $n$ for some polynomial $n$, and a word $w \in \Sigma^*$, we first construct a TM $\mathcal{M}_w$ that is started in on the empty tape and begins by writing $w$ onto the tape and then simulates $\mathcal{M}$ on it. This is a standard construction in complexity theory, and it is easy to see that the running time of $\mathcal{M}_w$ is bounded by a function

$2^{p'(|w|)}$ for some polynomial $p'$. With the observation made above, a computation of $\mathcal{M}_w$ can be seen as a sequence of configurations $C_1, \ldots, C_{p'(|w|)}$, with $|C_i| = i$. This does not directly define a tiling system, instead and again by a standard trick, cf. (van Emde Boas (1997)) or (Demri et al., 2016, Chp. 11), one compresses three adjacent tape cells into one tile in order to naturally derive a horizontal matching relation from overlaps between such triples and a vertical matching relation from the TM's transition function. At last, let $n' := p'(|w|)$. It is then a simple exercise to verify that a valid tiling of the triangle $\Delta_{n'}$ corresponds to an accepting run of $\mathcal{M}$ on $w$ and vice-versa, which establishes NEXPTIME-hardness.

(c) This is down exactly along the same lines as part (b), but instead making use of the fact that, when $n$ is given in unary encoding, $p(n)$ is polynomial in the size of the representation of $n$, and hence, the time needed for the guess-and-check procedure in the upper bound is only polynomial, and for the lower bound we need to assume that the running time of the TM is polynomially bounded. Thus, we get NP-completeness instead of NEXPTIME-completeness. $\qquad\square$

## B  PROOFS OF SECTION 4

In the following, we give formal proof for the undecidability results of Section 4. To do so, we make use of classical Feed-Forward Neural Networks.

**Feed-Forward Neural Network**  A *neuron* $v$ is a computational unit computing a function $\mathbb{R}^m \to \mathbb{R}$ by $v(x_1, \ldots, x_m) = \sigma(b + \sum_{i=1}^m w_i x_i)$ where $\sigma$ is a function called *activation* and $b, w_i$ are parameters called *bias* resp. *weight*. A *layer* $l$ is a tuple of nodes $(v_1, \ldots, v_n)$ where we assume that all nodes have the same input dimensionality $m$. Therefore, $l$ computes a function $\mathbb{R}^m \to \mathbb{R}^n$. We call $n$ the *size of layer* $l$. Let $l_1$ be a layer with input dimensionality $m$ and $l_k$ a layer of size $n$. A *Feed-Forward Neural Network (FNN)* $N$ is a tuple $(l_1, \ldots, l_k)$ of layers where we assume that for all $i \le k - 1$ holds that the size of $l_i$ equals the input dimensionality of $l_{i+1}$. Therefore, $N$ computes a function $\mathbb{R}^m \to \mathbb{R}^n$ by processing an input layer by layer.

In particular, we use specific FNN with $relu(x) = \max(0, x)$ activations, called *gadgets*, to derive lower bounds in connection with the expressibility of transformers. We denote the class of all FNN with $relu$ activations by $\mathcal{N}(relu)$.

**Lemma 4.** *Let $k \in \mathbb{R}^{>0}$. There are basic gadgets*

1. *$N_{|\cdot|} \in \mathcal{N}(relu)$ computing $N_{|\cdot|}(x) = |x|$,*

2. *$N_< \in \mathcal{N}(relu)$ computing a function $\mathbb{R}^2 \to \mathbb{R}$ such that $N_<(x_1, x_2) = 0$ if $(x_1 + 1) - x_2 \le 0$, $N_<(x_1, x_2) = (x_1 + 1) - x_2$ if $(x_1 + 1) - x_2 \in (0; 1)$ and $N_<(x_1, x_2) = 1$ otherwise,*

3. *$N_= \in \mathcal{N}(relu)$ computing a function $\mathbb{R}^2 \to \mathbb{R}$ such that $N_=(x_1, x_2) = 0$ if $x_1 - x_2 = 0$, $N_=(x_1, x_2) = |x_2 - x_1|$ if $|x_2 - x_1| \in (0; 1)$ and $N_=(x_1, x_2) = 1$ otherwise,*

4. *$N_\to \in \mathcal{N}(relu)$ computing a function $\mathbb{R}^2 \to \mathbb{R}$ such for all inputs $x_1, x_2$ with $x_1 \in \{0, 1\}$ and $x_2 \in [0; k]$ holds $N_\to(x_1, x_2) = 0$ if $x_1 = x_2 = 0$ or $x_1 = 1$ and $N_\to(x_1, x_2) = relu(x_2)$ otherwise.*

*Proof.* Let $N_{|\cdot|}$ be the minimal FNN computing $relu(relu(-x) + relu(x))$, let $N_<$ be the minimal FNN computing $relu(f_<(x_1, x_2) - f_<(x_1, x_2 + 1))$ where $f_<(y_1, y_2) = relu(y_1 - y_2 + 1)$ and let $N_=$ be the minimal FNN computing $relu(f_=(x_1, x_2) - f_=(x_1 + 1, x_2) + f_=(x_2, x_1) - f_=(x_2 + 1, x_1))$ where $f_=(y_1, y_2) = relu(y_2 - y_1)$. The claims of the lemma regarding these gadgets are straightforward given their functional form. Let $N_\to$ be the minimal FNN computing $relu(relu(x_2) - k \cdot relu(x_1))$. As stated in the lemma, we assume that $x_1 \in \{0, 1\}$ and $x_2 \in [0; k]$. Then, $-k \cdot relu(x_1)$ is $-k$ if $x_1 = 1$ and $0$ if $x_1 = 0$. Thus, $N_\to$ is guaranteed to be $0$ if $x_1 = 1$ and otherwise it depends on $x_2$. This gives the claim regarding gadget $N_\to$. $\qquad\square$

We will combine gadgets in different ways. Let $N_1$ and $N_2$ be FNN with the same input dimensionality $m$ and output dimensionality $n_1$ respectively $n_2$. We extend the computation of $N_1$ to functions $\mathbb{R}^{m'} \to \mathbb{R}^{n_1}$ with $m < m'$ by weighting additional dimensions with $0$ in the input layer. Given a set of input dimensions $x_1, \ldots, x_{m'}$, we denote the effective dimensions

$x_{i_1}, \ldots, x_{i_m}$ with pairwise different $i_j \in \{1, \ldots, m'\}$ by $N_1^{x_{i_1}, \ldots, x_{i_m}}$. Formally, this means that $N_1^{x_{i_1}, \ldots, x_{i_m}}(x_1, \ldots, x_{m'}) = N_1(x_{i_1}, \ldots, x_{i_m})$ for all inputs. We denote the FNN consisting of $N_1$ and $N_2$ placed next to each other by $N_1 \| N_2$. Formally, this is done by combining $N_1$ and $N_2$ layer by layer using 0 weights in intersecting connections. Then, $N_1 \| N_2$ computes $\mathbb{R}^m \to \mathbb{R}^{n_1+n_2}$ given by $N_1 \| N_2(\boldsymbol{x}) = (N_1(\boldsymbol{x}), N_2(\boldsymbol{x}))$. We generalize this operation to $k$ FNN $N_1 \| \cdots \| N_k$ in the obvious sense. Let $N_3$ be an FNN with input dimensionality $n_1$ and output dimensionality $n_3$. We denote the FNN consisting of $N_1$ and $N_3$ placed sequentially by $N_3 \circ N_1$. Formally, this is done by connecting the output layer of $N_1$ with the input layer of $N_3$. Then, $N_3 \circ N_1$ computes $\mathbb{R}^m \to \mathbb{R}^{n_3}$ given by $N_3 \circ N_1(\boldsymbol{x}) = N_3(N_1(\boldsymbol{x}))$.

We also consider specific gadgets needed in the context of tiling problems.

**Lemma 5.** *Let $S \subseteq \mathbb{N}$ be a finite set and $R \subseteq S^2$. There is FNN $N_R \in \mathcal{N}(relu)$ computing $\mathbb{R}^2 \to \mathbb{R}$ such that $N_R(x_1, x_2) \in \{0, 1\}$ if $(x_1, x_2) \in S^2$ and $N_R(x_1, x_2) = 0$ iff $(x_1, x_2) \in R$ and there is $N_{=t} \in \mathcal{N}(relu)$ for each $t \in S$ computing $\mathbb{R} \to \mathbb{R}$ such that $N_{=t}(x) \in \{0, 1\}$ for each $x \in \mathbb{N}$ and $N_{=t}(x) = 0$ iff $x = t$.*

*Proof.* Let $S \subseteq \mathbb{N}$ be finite, $R \subseteq S^2$ and $t \in S$. First, consider $N_{=t}$. Let $N_t$ be the minimal FNN computing $relu(0 \cdot x + t)$ and $N_{id}$ be the minimal FNN computing $(relu(x), -relu(-x))$. Obviously, $N_t$ computes the constant $t$ function and $N_{id}$ computes the identity in the form of two dimensional vectors. Let $N_{=t}$ be given by the minimal FNN computing $N_= \circ (N_{id} \| N_t)$ with the slight alteration that the two output dimensions of $N_{id}$ are connected to the first dimension of $N_=$. Then, the claim of the lemma regarding $N_{=t}$ follows from Lemma 4 and the operations on FNN described in Appendix B.

Now, consider $N_R$. Given some $s \in S$ let $R[s] = \{r \mid (s, r) \in R\}$. Let $N_\wedge^k$ be the minimal FNN computing $relu(x_1 + \cdots + x_k)$. Furthermore, let $N_{\in T}$ for some set $T \subseteq S$ be the minimal FNN such that $N_{\in T}(x) = 0$ if $x \in T$ and $N_{\in T}(x) = 1$ if $x \in S \setminus T$. A construction for $N_{\in T}$ is given in Theorem 4 in (Sälzer & Lange (2023)). According to this construction, $N_{\in T}$ consists of three layers and is polynomial in $T$. In the case that $T = \emptyset$ we assume that $N_{\in \emptyset}$ is the constant 1 function represented by a suitable FNN. Then, $N_R$ is given by $N_\wedge^{|S|} \circ ((N_\to \circ (N_{=s_1} \| N_{\in R[s_1]})) \| \cdots \| (N_\to \circ (N_{=s_{|S|}} \| N_{\in R[s_{|S|}]})))$ for some arbitrary order on $S$ with the slight alteration that $N_R$ has two input dimensions, meaning that each subnet $(N_{=s_i} \| N_{\in R[s_i]})$ is connected to the same two input dimensions. Again, the claim of the lemma regarding $N_R$ follows from Lemma 4 and the operations on FNN described in Appendix B. $\square$

Given these understandings of gadgets, we are set to formally prove the results of Section 4.

*Proof of Lemma 1.* Let $w = t_{0,0} t_{1,0} t_{1,1} t_{2,0} \cdots t_{m,n} \in S^+$ as stated in the lemma and assume some order $a_i$ on $S$. Furthermore, let $emb(a_i, 1) = (1, 1, 1, 1, i)$ and $emb(a_i, j) = (0, 1, j, \sum_{h=0}^j h, i)$ if $j > 1$. Let $emb(w) = \boldsymbol{x}_1^0 \cdots \boldsymbol{x}_k^0$. In the following, we build two layers $l_1$ and $l_2$ using components allowed in $\mathcal{T}_{udec}$, satisfying the statement of the lemma. Layer $l_1$ consists of a single attention head $att_{1,1} = (score_{1,1}, pool_{1,1})$. The scoring function is given by $score_{1,1}(\boldsymbol{x}_i^0, \boldsymbol{x}_j^0) = N_{1,1}(\langle Q_{1,1} \boldsymbol{x}_i^0, K_{1,1} \boldsymbol{x}_j^0 \rangle)$ where $Q_{1,1} = [(0, 0, -1, 0, 0), (0, 1, 0, 0, 0), (0, 1, 0, 0, 0)]$ and $K_{1,1} = [(0, 1, 0, 0, 0), (0, 1, 0, 0, 0), (0, 0, 0, 1, 0)]$ and $N(x) = -relu(x)$. We have that $score_{1,1}(\boldsymbol{x}_i^0, \boldsymbol{x}_j^0) = -relu((\sum_{h=0}^j h) - (i - 1))$ and it follows that $score_{1,1}(\boldsymbol{x}_i^0, \boldsymbol{x}_j^0) = 0$ if $\sum_{h=0}^j h \le i - 1$ and otherwise we have that $score_{1,1}(\boldsymbol{x}_i^0, \boldsymbol{x}_j^0) < 0$. The pooling function is specified by the matrix $W_{1,1} = [(1, 0, 0, 0, 0)]$ and uses $\text{hardmax}$ as normalisation function. The combination $comb_1$ function is given by the FNN $N_1(x_1, \ldots, x_5, y) = relu(x_2) \| \cdots \| relu(x_5) \| relu(y)$. Given a position $\boldsymbol{x}_i^0$, the attention head $att_{1,1}$ attends to all positions $\boldsymbol{x}_j^0$ satisfying $\sum_{h=0}^j h \le i - 1$. This is due to the way $score_{1,1}$ is build. Then, $att_{1,1}$ computes $\frac{1}{l}$ using $pool_{1,1}$ where $l$ is the number of positions $att_{1,1}$ attends to. Here, we exploit the fact that only the first position $\boldsymbol{x}_1^0$ has a non-zero entry in the its first dimension and that for all $i$ head $att_{1,1}$ attends to $\boldsymbol{x}_1^0$. Finally, $comb_1$ simply stacks the old vector $\boldsymbol{x}_i^0$ onto the value $\frac{1}{l}$, but leaves out the first dimension of $\boldsymbol{x}_i^0$. Let $l_1(emb(w)) = \boldsymbol{x}_1^1 \cdots \boldsymbol{x}_k^1$. Layer $l_2$ consists of a single attention head $att_{2,1} = (score_{2,1}, pool_{2,1})$. The scoring function $score_{2,1}$ is given by $N_{2,1}(\langle Q_{2,1} \boldsymbol{x}_i^1, K_{2,1} \boldsymbol{x}_j^1 \rangle)$ where $Q_{2,1} = [(0, 0, 0, 0, 1)]$, $K_{2,1} = [(0, 1, 0, 0, 0)]$ and $N_{2,1}(x) = -relu(relu(x - 1) + relu(1 - x))$. We have that $score_{2,1}(\boldsymbol{x}_i^1, \boldsymbol{x}_j^1) = 0$ if $\frac{1}{l} \cdot j = 1$ where

$\frac{1}{l}$ is the fifth dimension of $\boldsymbol{x}_i^1$ and otherwise $score_{2,1}(\boldsymbol{x}_i^1, \boldsymbol{x}_j^1) < 0$. The pooling function $pool_{2,1}$ is specified by $W_{2,1} = [(0,1,0,0,0), (0,0,1,0,0)]$ and uses $\mathrm{hardmax}$ as normalisation. The combination $comb_2$ is given by the FNN $N_2(x_1, \ldots, x_5, y_1, y_2) = relu(x_1)\|relu(x_2)\|relu(y_1)\|relu(x_2 - y_2 - 1)\|relu(x_4)$. Given a position $\boldsymbol{x}_i^1$, the attention head $att_{2,1}$ attends to the position $j$, where $\frac{1}{l} \cdot j = 1$. Relying on our arguments regarding the computation of $l_1$, this is the position $j$ satisfying $\max_j(\sum_{h=0}^{j} h \leq i - 1)$. However, this $j$ is equal to the row index $r(i)$ of the decomposition of $i$ based on the inversion of Cantor's pairing function. Thus, we have that $r(i) = j$. Furthermore, we have that $c(i) = (i - 1) - (\sum_{h=0}^{j} h)$, which is computed by $relu(x_2 - y_2 - 1)$ in the combination function $comb_2$. Overall, we see that $l_2(l_1(emb(w)))$ gives the desired result. $\square$

*Proof of Lemma 2.* Let $f$ be as stated in the lemma. By definition of $\mathcal{T}_{udec}$, the scoring function of $att_f$ is of the form $N(\langle Q\boldsymbol{x}_i, K\boldsymbol{x}_j\rangle)$ and the normalisation is $\mathrm{hardmax}$. Let $Q = [(a_1, \ldots, a_k), (b, 0, \ldots, 0), (1, 0, \ldots, 0)]$, $K = [(1, 0, \ldots, 0), (1, 0, \ldots, 0), (0, -1, 0, \ldots, 0)]$ and $N$ be the minimal FNN computing $N(x) = -relu(N_{|\cdot|}(x)) = -|x|$ where $N_{|\cdot|}$ is given by Lemma 4. Overall, this ensures that the scoring is given by $score(\boldsymbol{x}_i, \boldsymbol{x}_j) = -|f(\boldsymbol{x}_i) - j|$. Then, the statement of the lemma follows from the fact that $\mathrm{hardmax}$ attends to the maximum, which is 0 given this scoring, and that $j \in \mathbb{N}$ is unique for each $\boldsymbol{x}_j$. $\square$

**Lemma 6.** *There is attention head $att_\leq$ in $\mathcal{T}_{udec}$ such that for all sequences $\boldsymbol{x}_1, \ldots, \boldsymbol{x}_m$ where all $\boldsymbol{x}_i = (1, i, \boldsymbol{y}_i)$ the head $att_\leq$ attends to $\{\boldsymbol{x}_1, \ldots, \boldsymbol{x}_i\}$ given $i$.*

*Proof.* By definition of $\mathcal{T}_{udec}$, the scoring function of $att_f$ is of the form $N(\langle Q\boldsymbol{x}_i, K\boldsymbol{x}_j\rangle)$ and the normalisation is $\mathrm{hardmax}$. Let $Q = [(0, 1, 0, \ldots, 0), (1, 0, \ldots, 0)]$ and let $K$ be equal to $[(1, 0, \ldots, 0), (0, -1, 0, \ldots, 0)]$. Furthermore, let $N(x) = -relu(x)$. We observe that $N$ outputs 0 if $j \leq i$ and otherwise $N(x) < 0$. In combination with $\mathrm{hardmax}$, this ensures that $att_\leq$ behaves as stated by the lemma. $\square$

*Proof of Theorem 1.* We prove the statement via reduction from OTWP$^*$. Let $\mathcal{S} = (S, H, V, t_I, t_F)$ be an instance of OTWP$^*$ with $|S| = k$. W.l.o.g we assume that $S \subseteq \mathbb{N}$. Let $T_\mathcal{S} \in \mathcal{T}_{udec}$ built the following way. $T_\mathcal{S}$ uses the embedding $emb$ of transformer in $\mathcal{T}_{udec}$ specified in the beginning of Section 4. Furthermore, it has four layers. Layers $l_1$, $l_2$ are as in Lemma 1. Layer $l_3$ is given by $l_3 = (att_{prev}, att_{next}, att_{step}, comb_3)$ where $att_{prev}$, $att_{next}$ and $att_{step}$ are of Lemma 2 whereby $prev(x_1, \ldots, x_5) = x_2 - 1$, $next(x_1, \ldots, x_5) = x_2 + 1$ and $step(x_1, \ldots, x_5) = x_2 + x_3 + 1$. We assume that all three attention heads use the identity matrix as linear maps in their respective pooling function. $comb_3$ is given by an FNN $N_3$ computing $\mathbb{R}^{4 \cdot 5} \to \mathbb{R}$. Let the input dimensions of $N_3$ be $x_{1,1}, \ldots, x_{1,5}, x_{2,1}, \ldots, x_{4,5}$. Then, $N_3$ is equal to

$$relu(x_{1,1})\|relu(x_{1,2})\|N_a\|N_{b_1}\|N_{b_2}\|N_c\|N_d$$

where $N_a = N_\rightarrow \circ (N_=^{x_{1,2}, x_{3,2}}\|N_=^{x_{1,3}, x_{1,4}})$, $N_{b_1} = N_\rightarrow \circ (N_=^{x_{1,2}, x_{2,2}}\|N_{=t_I}^{x_{1,5}})$, $N_{b_2} = N_\rightarrow \circ (N_=^{x_{1,2}, x_{3,2}}\|N_{=t_F}^{x_{1,5}})$, $N_c = N_\rightarrow \circ (N_<^{x_{1,4}, x_{1,3}}\|N_H^{x_{1,5}, x_{3,5}})$ and $N_d = N_\rightarrow \circ (N_<^{x_{1,3}, x_{4,3}}\|N_V^{x_{1,5}, x_{4,5}})$ using the gadgets and constructions described in Appendix B. Layer $l_4$ is given by $l_4 = (att_{leq}, comb_4)$ where $att_{leq}$ attends to $\{\boldsymbol{x}_1, \ldots, \boldsymbol{x}_i\}$ given $i$ and $comb_4$ is given by the minimal FNN $N_4$ computing $relu(x_3 + \cdots + x_7)$. A formal proof for the existence of $att_{leq}$ in $\mathcal{T}_{udec}$ is given in Lemma 6. Furthermore, the output function $out$ of $T_\mathcal{S}$ is given by the minimal FNN $N_{out}$ computing $N(x_1) = relu(1 - x_1)$.

Let $w = t_1 \cdots t_l \in S^*$ be some word over alphabet $S$. As defined above, we have that $emb(t_i, i) = (1, i, \sum_{j=0}^{i} j, k_i)$ where $k_i \in \{1, \ldots, |S|\}$. Consider $\boldsymbol{x}_1^2 \cdots \boldsymbol{x}_m^2$, namely the sequence of vectors after propagating $w$ through the embedding $emb$ and layers $l_1, l_2$ of $T_\mathcal{S}$. As stated by Lemma 1, we have that $\boldsymbol{x}_i^2 = (1, i, r(i), c(i), k_i)$ where $r(i)$ and $c(i)$ are the row respectively column of tile $t_i$ if we interpret $w$ as an encoded tiling. Note that all vectors $\boldsymbol{x}_i^3$ are non-negative due to the way $N_3$ is built. In the following, we argue that all $\boldsymbol{x}_i^3 = \boldsymbol{0}$ if and only if $w$ is a valid encoded tiling. Given this equivalence, the statement of the lemma follows immediately as $l_4$ simply sums up all vectors and dimensions (except for the first and second) of $\boldsymbol{x}_1^3, \ldots, \boldsymbol{x}_m^3$ in $\boldsymbol{x}_m^4$ and the output of $N_4$ indicates whether there was some non-zero value. We fix some arbitrary $\boldsymbol{x}_i^2 = (1, i, r(i), c(i), k_i)$. Then, $\boldsymbol{x}_i^3 = N_3(\boldsymbol{x}_i^2, \boldsymbol{x}_{i_{prev}}^2, \boldsymbol{x}_{i_{next}}^2, \boldsymbol{x}_{i_{step}}^2)$ where $i_{next} = i + 1$ if $i < m$ and $m$ otherwise, $i_{prev} = i - 1$ if $i > 1$ and 1 otherwise and $i_{step} = i + r(i) + 1$ if $i < m - r(i) - 1$ and $m$ otherwise.

Consider property $(a)$ and subnetwork $N_a$. With the understanding gained in Appendix B, $N_{=}^{x_{1,2},x_{3,2}}$ outputs 0 iff $x_{1,2} = x_{3,2}$. These dimensions correspond to positions $i$ and $i_{next}$, which are only equal if $i = m$ (Lemma 2). Furthermore, the property of $N_{\rightarrow}$ stated by Lemma 4 is given as the output of $N_{=}$ is guaranteed to be in $[0;1]$ and the values of $x_{1,2}$ and $x_{3,2}$ are guaranteed to be in $\mathbb{N}$. In summary, this ensures that the third dimension of $\boldsymbol{x}_m^3$ is 0 iff $r(m) = c(m)$. For other positions the third dimension is always 0 since $N_{\rightarrow}$ outputs 0 in these cases due to the fact that $N_{=}^{x_{1,2},x_{3,2}}$ equals 1. Analogously, $N_{b_1}$ and $N_{b_2}$ ensure that $t_1 = t_I$ and $t_m = t_F$ and, thus, property (b) iff the fourth and fifth dimensions in all positions are equal to 0. Consider properties (c) and (d) described above and assume that property (a) holds. These two properties are non-local in the sense that they depend on at least two positions in $\boldsymbol{x}_1^2 \cdots \boldsymbol{x}_m^2$. Consider the subnet $N_c$. By construction and the gadgets described in Appendix B, we have that $N_c$ outputs 0 if $c(i) < r(i)$ and $(t_i, t_{i+1}) \in H$ or if $c(i) = r(i)$, which means that tile $t_i$ is rightmost in its corresponding row. Otherwise the value computed by $N_c$ is greater than 0. Analogously, subnet $N_d$ checks whether vertically stacked tiles do match. In summary, this ensures that the sixth and seventh dimension of each $\boldsymbol{x}_i^3$ is equal to 0 if and only if properties (c) and (d) hold. □

*Proof of Theorem 2.* In the same manner as in the proof of Theorem 1, we prove the statement via reduction from OTWP$^*$. The reduction is exactly the same, namely given an OTWP$^*$ instance $\mathcal{S} = (S, H, V, t_I, t_F)$ we build TE $T_{\mathcal{S}}$ which recognizes exactly those words $w$ representing a valid encoded tiling of $\mathcal{S}$. For details, see the proof of Theorem 1.

Given the correctness arguments for $T_{\mathcal{S}}$ in Theorem 1, it is left to argue that $T_{\mathcal{S}}$ works as intended, despite the fact that it works over some FA $F$ using at most $\mathcal{O}(\log(\max(|S|, n)))$ bits where $n$ is the length of an input word. We choose $F$ such that overflow situations do not occur in any computation $T_{\mathcal{S}}(w)$ and rounding is handled such that $T_{\mathcal{S}}$ works as intended. Throughout this proof, we use $\log(n)$ Namely, given a word $w$ with $|w| = n$ assume that $F$ uses $m = \lfloor 4\log(\max(|S|, n)) \rfloor + 2$ bits and rounds values off to the nearest representable number. We denote the value resulting from rounding $x$ off in arithmetic $F$ by $\lfloor x \rfloor_F$. We assume that there is an extra bit that is used as a sign bit and that at least $\lfloor 3\log(n) \rfloor + 1$ bits can be used to represent integer and at least $\lfloor \log(n) \rfloor + 1$ bits can be used to represent fractional parts. Note that this is a reasonable assumption for all common FA, like fixed-point or floating-point arithmetic. Furthermore, it is clearly the case that $m \in \mathcal{O}(\log(\max(|S|, n)))$. To ease our arguments and notation from here on, we assume w.l.o.g. that we represent $n$ using $\log(n)$ instead of $\lfloor log(n) \rfloor + 1$.

Per definition, $T_{\mathcal{S}}$ uses the embedding function $emb(a_k, 0) = (1, 1, 0, 0, k)$ and $emb(a_k, i) = (0, 1, i, \sum_{j=0}^{i} j, k)$. First, we assume that each $k$, namely the value representing a specific tile from $S$, is a unique, positive value. This is possible as $F$ uses $m > \log(|S|)$ bits. Furthermore, we see that $emb$, especially the sum $\sum_{j=0}^{i} j = \frac{i(i+1)}{2} \leq i^2$, works as intended up to $i = n$ due to the fact that $F$ uses more than $m > 2\log(n)$ bits to represent integer parts. Next, consider layer $l_1$ and $l_2$ of Lemma 1. Layer $l_1$ consists of a single attention head $att_{1,1}$. Here, the only crucial parts are the computation of value $\frac{1}{l}$ in $pool_{1,1}$ for a position $i$. Per definition, $l$ corresponds to the number of positions $j$ such that $\sum_{h=0}^{j} h \leq i - 1$. As $i$ is bounded by $n$, this inequality can only be satisfied by positions $j$ for which $j \leq \sqrt{n}$ holds. As $T_{\mathcal{S}}$ uses $\mathrm{hardmax}$ to count the positions for which this inequality holds, $l$ is bounded by $\sqrt{n}$. Next, we observe that $\lfloor \frac{1}{l} \rfloor_F = \frac{\lfloor 2^{\log(n)} \frac{1}{l} \rfloor}{2^{\log(n)}} = \frac{\lfloor \frac{n}{l} \rfloor}{n}$, namely the general understanding of rounding off where we use $\log(n)$ bits to represent fractions. However, this gives that for all $1 \leq l_1 < l_2 \leq \sqrt{n}$ that $\lfloor \frac{1}{l_1} \rfloor_F \neq \lfloor \frac{1}{l_2} \rfloor_F$ as $\lfloor \frac{n}{l_1} \rfloor \neq \lfloor \frac{n}{l_2} \rfloor$ holds for all $l_1 < l_2 \leq \sqrt{n}$. This means, that it is ensured by $F$ that $\frac{1}{l}$ is uniquely representable.

Next, the only crucial part in $l_2$ is the computation of the product $\frac{1}{l} \cdot j$, which is used to determine the position $j$ for which $\frac{1}{l} \cdot j = 1$ in $score_{2,1}$, which is obviously given by position $l$. This equality is no longer guaranteed to exist if we consider $\lfloor \frac{1}{l} \rfloor_F \cdot j$. However, due to the monotonicity of $\lfloor \frac{1}{l} \rfloor_F$ for $l \leq \sqrt{n}$ and that the maximum round of error is given by $\frac{1}{2^{\log(n)}}$, we have that the $j = l$ produces the value closest to 1 in the product $\frac{1}{l} \cdot j$. Taking a look at $score_{2,1}$, this ensures that $l$ is still the position that $att_{2,1}$ attends to. Therefore, the statement of Lemma 1 is still valid for $T_{\mathcal{S}}$ working over $F$. We observe that all values of some vector $\boldsymbol{x}_j^2$ after layer $l_2$ are positive integers whose magnitude is bounded by $n^2$.

Now, consider layer $l_3$ and $l_4$. From the proof of Theorem 1 we see that the gadgets at most sum up two values or compute a fraction of the form $\frac{i+j}{2}$ and $\frac{i-j}{2}$ (in gadgets $N_H$ or $N_V$). Both can safely be done with at least $3\log(n)$ bits for integer and $\log(n)$ for fractional parts, as all previously computed values, up to layer $l_2$, in a computation of $T_{\mathcal{S}}(w)$ are representable using $2\log(n)$ bits. We observe that the values of the third to seventh dimension of some $\boldsymbol{x}_j^3$ are either 0 or 1. This is due to the fact that all values after layer $l_2$ are guaranteed to be integers. Next, consider layer $l_4$. The computation done by $att_\leq$ is safe (see Lemma 6) and the crucial step here is the computation of $comb_4$ given by $relu(x_3 + \cdots + x_7)$. The values $x_i$ are all of the form $\frac{i}{j}$ where $i$ is guaranteed to be 0 or 1 and $j$ is the normalisation induced by $att_\leq$ from perspective of position $j$. However, this means $j$ is bounded by $n$ and, thus, $\lfloor \frac{i}{j} \rfloor_F > 0$ if and only if $i = 1$ for all $j$ due to the fact that $F$ allows for $\log(n)$ bits to represent fractional parts. Finally, $out$ is trivially computable in $F$, which finishes the proof. $\square$

## C    PROOFS OF SECTION 5

*Proof of Theorem 3.* The decidability and membership results of statements (1) and (2 )are sufficiently argued in the proof sketch given in Section 5.

To prove the hardness results of statements (1) and (2), we establish a reduction from OTWP$_{\mathsf{un}}$ respectively OTWP$_{\mathsf{bin}}$: given some bounded word-tiling instance $(\mathcal{S}, n)$ we build an instance $(T_{\mathcal{S}}, n)$ of BTRSAT$_{\mathsf{un}}$ respectively BTRSAT$_{\mathsf{bin}}$ where $T_{\mathcal{S}}$ is build as described in Theorem 1. The only missing argument is that these reductions are polynomial. In particular, this means that $T_{\mathcal{S}}$ must be built in polynomial time regarding the size of $(\mathcal{S}, n)$. Therefore, we recall the proof of Theorem 1.

First, we see that the embedding function $emb$ and the amount of layers of $T_{\mathcal{S}}$ is independent of $\mathcal{S}$ and $n$. The first two layers $l_1$ and $l_2$ of $T_{\mathcal{S}}$ are specified in Lemma 1. Recalling the proof of Lemma 1, we see that $l_1$ and $l_2$ each consist of a single attention head, whose internal parameters like scoring, pooling or combination are independent of $(\mathcal{S}, n)$ as well. Next, consider layer $l_3$. This layer consists of three attention heads $att_{prev}$, $att_{next}$ and $att_{step}$ each given by the template described in Lemma 2, which again is independent of $(\mathcal{S}, n)$. Additionally, $l_3$ contains the combination function $comb_3$. This combination function is represented by a FNN $N_3$, using smaller FNN $N_a$, $N_{b_1}$, $N_{b_2}$, $N_c$ and $N_d$ as building blocks. These are dependent on $\mathcal{S}$, as they are built using gadgets $N_{=t_I}$, $N_{=t_F}$, $N_H$ and $N_V$ where $t_I$, $t_F$, $H$ and $V$ are components of $\mathcal{S}$. However, in the proof of Lemma 5 we see that these gadgets are at most polynomial in their respective parameter. Layer $l_4$ and the output function, specified by FNN $N_{out}$, are again independent of $(\mathcal{S}, n)$. In summary, the TE $T_{\mathcal{S}}$ is polynomial in $(\mathcal{S}, n)$, which makes the reductions from OTWP$^{\mathsf{exp}}$ und OTWP$^{\mathsf{poly}}$ polynomial. $\square$

Next, we address the proof of Lemma 3. We need some preliminary, rather technical result first. Let $T$ be an TE and $w \in \Sigma^+$ be a word and consider the computation $T(w)$. Let $X_{T(w)}^0 = emb(w)$ and $X_{T(w)}^i$ be the sequence of vectors occurring after the computation of layer $l_i$ of $T$. Let $\boldsymbol{x}$ and $\boldsymbol{x}'$ be two vectors matching the dimensionality of $score_{i,j}$ of $T$. Overloading some notation, let $N_w(\boldsymbol{x}, \boldsymbol{x}', i, j) = norm_{i,j}(score_{i,j}(\boldsymbol{x}, \boldsymbol{x}'), score_{i,j}(\boldsymbol{x}, X_{T(w)}^{i-1}))$ where $score_{i,j}(\boldsymbol{x}, X_{T(w)}^{i-1})$ is the vector of all scorings of $\boldsymbol{x}$ with sequence $X_{T(w)}^{i-1}$. We remark that it is not necessary that $\boldsymbol{x}$ or $\boldsymbol{x}'$ must occur in $X_{T(w)}^{i-1}$ for this to be well defined. Again overloading some notation, let $P_w(\boldsymbol{x}, i, j) = pool_{i,j}(X_{T(w)}^{i-1}, score_{i,j}(\boldsymbol{x}, X_{T(w)}^{i-1}))$.

**Lemma 7.** *Let $T$ be a additive-periodical TE of depth $L$, maximum width $H$ and periodicity $p$ with $norm_{i,j} \in \{\text{softmax}, \text{hardmax}\}$ for all $i \leq L, j \leq H$, let $w = u_1 u_{j_1} \cdots u_{j_n} u_2 \in \Sigma^+$ where $u_1, u_2 \in \Sigma^+$, all $u_{j_i} \in \Sigma^p$ and all $u_{j_i}$ also occur in $u_1$ or $u_2$ and let $\mathcal{X}$ be the set of all vectors occurring in any of the sequences $X_{T(w)}^i$. If there are indexes $h_1 < h_2 \leq h$ such that for all $\boldsymbol{x}, \boldsymbol{x}' \in \mathcal{X}, i \leq L, j \leq H$ holds that $N_{u_1 u_{j_1} \cdots u_{j_{h_1}}}(\boldsymbol{x}, \boldsymbol{x}', i, j) = N_{u_1 u_{j_1} \cdots u_{j_{h_2}}}(\boldsymbol{x}, \boldsymbol{x}', i, j)$ and $P_{u_1 u_{j_1} \cdots u_{j_{h_1}}}(\boldsymbol{x}, i, j) = P_{u_1 u_{j_1} \cdots u_{j_{h_2}}}(\boldsymbol{x}, i, j)$ then it holds that $N_{u_1 u_{j_1} \cdots u_{j_{h_1}} u_{j_{h_2+1}} \cdots u_2}(\boldsymbol{x}, \boldsymbol{x}', i, j) = N_{u_1 \cdots u_2}(\boldsymbol{x}, \boldsymbol{x}', i, j)$ and $P_{u_1 u_{j_1} \cdots u_{j_{h_1}} u_{j_{h_2+1}} \cdots u_2}(\boldsymbol{x}, i, j) = P_{u_1 \cdots u_2}(\boldsymbol{x}, i, j)$.*

*Proof.* Let $T$, $w$, $\mathcal{X}$, $h_1$ and $h_2$ be as stated above. We prove the statement via induction on the layers $l_i$. First, consider layer $l_1$ and fix some tuple $(\boldsymbol{x}, \boldsymbol{x}', 1, j)$. We first show that $N_{u_1 u_{j_1} \cdots u_{j_{h_1}} u_{j_{h_2}+1} \cdots u_2}(\boldsymbol{x}, \boldsymbol{x}', 1, j) = N_{u_1 \cdots u_2}(\boldsymbol{x}, \boldsymbol{x}', 1, j)$. Assume that $norm_{1,j}$ is given by softmax. Then, $norm_{1,j}$ computes $\frac{e^{score_{1,j}(\boldsymbol{x},\boldsymbol{x}')}}{\sum_{score_{1,j}(\boldsymbol{x}, X^0_{T(w')})} e^{s_{i'}}}$ for all words $w'$. Obviously, the numerator in $N_{u_1 \cdots u_{h_1} u_{h_2}+1 \cdots u_2}(\boldsymbol{x}, \boldsymbol{x}', 1, j)$ and $N_{u_1 \cdots u_2}(\boldsymbol{x}, \boldsymbol{x}', 1, j)$ is equal. By definition, we have that $score_{i,j}$ is local in the sense that it compares vectors pairwise, producing the different scoring values $s_{i'}$ independent of the overall word. Furthermore, due to the fact that $emb$ is additive-periodical, we have $X^0_{T(u_1 u_{j_1} \cdots u_{j_{h_1}} u_{j_{h_2}+1} \cdots u_2)}$ and $X^0_{T(u_1 \cdots u_2)}$ are equal in the sense that the vectors corresponding to $u_{j_{h_2}+1} \cdots u_2$ are equal. We refer to this property (*) later on. Using these observations and that $N_{u_1 u_{j_1} \cdots u_{j_{h_1}}}(\boldsymbol{x}, \boldsymbol{x}', 1, j) = N_{u_1 u_{j_1} \cdots u_{j_{h_2}}}(\boldsymbol{x}, \boldsymbol{x}', 1, j)$, we have that the denominator is equal as well. Now, assume that $norm_{1,j}$ is given by hardmax. Then, $norm_{1,j}$ computes $\frac{f(score_{1,j}(\boldsymbol{x},\boldsymbol{x}'), score_{1,j}(\boldsymbol{x}, X^0_{T(w')}))}{\sum_{score_{1,j}(\boldsymbol{x}, X^0_{T(w')})} f(s_{i'}, score_{1,j}(\boldsymbol{x}, X^0_{T(w')}))}$ where $f(s, S) = 1$ if $s$ is maximal in $S$ and 0 otherwise for any word $w'$. In contrast to softmax, we have that the values of $f(\cdots)$ are dependent of the overall context, namely the vector of all scorings $score_{1,j}(\boldsymbol{x}, X^0_{T(w')})$. Compare $X^0_{T(u_1 u_{j_1} \cdots u_{j_{h_1}} u_{j_{h_2}+1} \cdots u_2)}$ and $X^0_{T(u_1 \cdots u_2)}$, both given by the additive-periodical embedding $emb$. Via assumption, we have that each $u_{j_i}$ block also occurs in $u_1$ or $u_2$. In particular, this means every vector that occurs in $emb(u_1 \cdots u_2)$ does also occur in $emb(u_1 u_{j_1} \cdots u_{j_{h_1}} u_{j_{h_2}+1} \cdots u_2)$ and vice-versa. This implies that $f(score_{1,j}(\boldsymbol{x}, \boldsymbol{x}'), score_{1,j}(\boldsymbol{x}, X^0_{T(u_1 u_{j_1} \cdots u_{j_{h_1}} u_{j_{h_2}+1} \cdots u_2)})) = f(score_{1,j}(\boldsymbol{x}, \boldsymbol{x}'), score_{1,j}(\boldsymbol{x}, X^0_{T(u_1 \cdots u_2)}))$ for any scoring value $score_{1,j}(\boldsymbol{x}, \boldsymbol{x}')$. In combination with the assumption that $N_{u_1 u_{j_1} \cdots u_{j_{h_1}}}(\boldsymbol{x}, \boldsymbol{x}', 1, j) = N_{u_1 u_{j_1} \cdots u_{j_{h_2}}}(\boldsymbol{x}, \boldsymbol{x}', 1, j)$ and the observations above, we also get $N_{u_1 u_{j_1} \cdots u_{j_{h_1}} u_{j_{h_2}+1} \cdots u_2}(\boldsymbol{x}, \boldsymbol{x}', 1, j) = N_{u_1 \cdots u_2}(\boldsymbol{x}, \boldsymbol{x}', 1, j)$ in the hardmax case. Next, consider the pooling functions. By definition, we have that $pool_{1,j}(X^0_{T(w')}, score_{1,j}(\boldsymbol{x}, X^0_{T(w')}))$ computes $\sum_{X^0_{T(w')}} norm_{1,j}(\boldsymbol{x}, \boldsymbol{x}_{i'}, score_{i,j}(\boldsymbol{x}, X^0_{T(w')}))(W\boldsymbol{x}_{i'})$ for any word $w'$. Our previous arguments give that $N_{u_1 u_{j_1} \cdots u_{j_{h_1}} u_{j_{h_2}+1} \cdots u_2}(\boldsymbol{x}, \boldsymbol{x}', 1, j) = N_{u_1 \cdots u_2}(\boldsymbol{x}, \boldsymbol{x}', 1, j)$. In combination with $P_{u_1 u_{j_1} \cdots u_{j_{h_1}}}(\boldsymbol{x}, i, j) = P_{u_1 u_{j_1} \cdots u_{j_{h_2}}}(\boldsymbol{x}, i, j)$ and (*), we immediately get that $P_{u_1 u_{j_1} \cdots u_{j_{h_1}} u_{j_{h_2}+1} \cdots u_2}(\boldsymbol{x}, i, j) = P_{u_1 \cdots u_2}(\boldsymbol{x}, i, j)$ holds as well. Next, consider layer $l_i$. The arguments are exactly the same as in the base case. However, we need to rely on the induction hypothesis. Namely, we assume that all $pool_{i-1,j}$ produce the same output in computation $T(u_1 u_{j_1} \cdots u_{j_{h_1}} u_{j_{h_2}+1} \cdots u_2)$ and computation $T(u_1 \cdots u_2)$. This implies that all vectors present in $X^{i-1}_{T(u_1 u_{j_1} \cdots u_{j_{h_1}} u_{j_{h_2}+1} \cdots u_2)}$ are also present in $X^{i-1}_{T(u_1 \cdots u_2)}$ and vice-versa and that the vectors corresponding to $u_{j_{h_2}+1} \cdots u_2$ are equal in both computations. $\square$

*Proof of Lemma 3.* Let $T \in \mathcal{T}^{\text{FIX}}_{\circ}$ be an additive-periodical TE working over alphabet $\Sigma$, having periodicity $p$, depth $L$, maximum width $H$, maximum dimensionality $D$ and working over an FA $F$ using $b$ bits for binary encoding. We use $V$ to denote the set of values representable in the fixed arithmetic that $T$ works over. Note that $|V| \leq 2^b$. Let $w \in \Sigma^+$ be a word such that $T(w) = 1$. We observe that there is $m \in \mathbb{N}$ such that $w = u_1 \cdots u_m u$ where $u_i \in \Sigma^p$ are blocks of symbols of length $p$ and $u \in \Sigma^{\leq p}$. Our goal is to prove that a not necessarily connected subsequence of at most $2^{(|T|)^6}$ many $p$-blocks $u_i$ from $u_1 \cdots u_m$ is sufficient to ensure the same computation of $T$. In the case that $pm + p \leq 2^{(|T|)^6}$ we are done. Therefore, assume that $m > 2^{(|T|)^6}$.

Let $U$ be the set of all unique $u_i$. We observe that $|U| \leq |\Sigma|^p$. Next, we fix some not necessarily connected but ordered subsequence $S = u_{j_0} u_{j_1} \cdots u_{j_n} u_{j_{n+1}}$ with $u_{j_0} = u_1$, $j_i \in \{2, \ldots, m\}$ and $u_{j_{n+1}} = u$ of $w$ such that each $u' \in U$ occurs exactly once. For the case that $u_1 = u$ we allow this specific block to occur twice in $S$. The assumption $m > 2^{poly(|T|)}$ implies that $S \neq w$. This means that there are pairs $(u_{j_h}, u_{j_{h+1}})$ in $S$ with some non-empty sequence of $p$-blocks $u_{j'_1} \cdots u_{j'_l}$ in between. W.lo.g. assume $u_{j_0}$ and $u_{j_1}$ is such a pair. Our goal is to argue that there are at most $2^{(|T|)^5}$ blocks from $u_{j'_1} \cdots u_{j'_l}$ needed to ensure the same computation of $T$. Given that this argument works for all $|\Sigma|^p$ adjacent pairs in $S$, we are done.

Consider the computation $T(w)$. The additive-periodical embedding $emb$ of $T$ implies that $emb(w)$ includes at most $\Sigma p$ different vectors. Furthermore, from layer to layer equal vectors are mapped equally, which means that each $X_w^1, \ldots, X_w^L$ contains at most $\Sigma p$ different vectors as well. This implies that the computation $T(w)$ induces at most $(L\Sigma p)^2 \times L \times H \leq (\Sigma p L^2 H)^2 \leq (\Sigma p L H)^4$ different tuples $(\boldsymbol{x}, \boldsymbol{x}', i, j)$ where $\boldsymbol{x}, \boldsymbol{x}'$ are vectors induced by $T(w)$ and $i \leq L, j \leq H$. Additionally, we have that for each value $N_w(\boldsymbol{x}, \boldsymbol{x}', i, j)$ and $P_w(\boldsymbol{x}, i, j)$, as defined in the beginning of this section, there are at most $|V^D| \leq 2^{bD}$ possibilities. Simple combinatorics, namely the pigeon hole principle, states that in the increasing sequence $u_{j_1'}, u_{j_2'}, \ldots$ there must be points $h_1$ and $h_2$ with $h_1 \leq 2^{bD(\Sigma p L H)^4} \leq 2^{(|T|)^5}$ such that for all tuples $(\boldsymbol{x}, \boldsymbol{x}', i, j)$ induced by $T(w)$ we have that $N_{u_{j_0} u_{j_1'} \cdots u_{j_{h_1}'}}(\boldsymbol{x}, \boldsymbol{x}', i, j) = N_{u_{j_0} u_{j_1'} \cdots u_{j_{h_2}'}}(\boldsymbol{x}, \boldsymbol{x}', i, j)$ and $P_{u_{j_0} u_{j_1'} \cdots u_{j_{h_1}'}}(\boldsymbol{x}, i, j) = P_{u_{j_0} u_{j_1'} \cdots u_{j_{h_2}'}}(\boldsymbol{x}, i, j)$. Now, Lemma 7 states that this implies $N_{u_{j_0} u_{j_1'} \cdots u_{j_{h_1}'} u_{j_{h_2+1}'} \cdots u_{j_1} \cdots u}(\boldsymbol{x}, \boldsymbol{x}', i, j) = N_w(\boldsymbol{x}, \boldsymbol{x}', i, j)$ and $P_{u_{j_0} u_{j_1'} \cdots u_{j_{h_1}'} u_{j_{h_2+1}'} \cdots u_{j_1} \cdots u}(\boldsymbol{x}, i, j) = P_w(\boldsymbol{x}, i, j)$. However, this implies that the subsequence $u_{j_{h_1+1}'} \cdots u_{j_{h_2}'}$ has no influence in the computation of $T$ on $w$ and, thus, can be left out. As we can argue this for every such cycle occurring in $u_{j_1'} \cdots u_{j_l'}$, we get the desired bound of $2^{(|T|)^5}$. $\qquad\square$

*Proof of Theorem 5.* First, we argue the decidability of TRSAT[$\mathcal{T}^{\text{FIX}}$]. Assume that $T \in \mathcal{T}^{\text{FIX}}$ with an arbitrary embedding $emb$ is given that operates in a fixed-width arithmetic using $b$ bits for representing numbers and wrap-around to handle overflow. Then, $emb$ is periodic with periodicity $p \leq 2^b$, simply due to the fact that positions $i$ in some word $w$ can only be exactly represented up to magnitude $2^b$. Therefore, the same arguments as used in Theorem 4 apply here. Note that this does not imply NEXPTIME-membership of TRSAT[$\mathcal{T}^{\text{FIX}}$], due to the fact that the period is exponential in $b$. Analogously, in a saturating scenario, we have that $emb$ has a finite prefix of length at most $2^b$ and is periodic with periodicity 1 afterwards. Here, the small-word property used in Theorem 4 follows the same line of reasoning, with the difference that either the finite prefix is sufficient as a witness, or the finite prefix followed by an exponentially bounded suffix, whose existence follows from the same arguments as in Lemma 3 with periodicity $p = 1$.

Second, we argue the NEXPTIME-hardness. We prove the statement via reduction from OTWP$_{\text{bin}}$. Let $\mathcal{S} = (S, H, V, t_I, t_F)$ and $n \geq 1$ be an instance of OTWP$_{\text{bin}}$. We construct an TE $T_{\mathcal{S},n} \in \mathcal{T}^{\text{FIX}}$ working over some FA $F$ with $T_{\mathcal{S},n}(w) = 1$ if and only if $w \in S^+$ witnesses the validity of the OTWP$_{\text{bin}}$ instance $(\mathcal{S}, n)$.

Next, let $T_{\mathcal{S},n}$ be built exactly like $T_\mathcal{S}$ in the proof of Theorem 4, but with the following structural adjustments. In layer $l_3$ we adjust $comb_3$ to be $comb_3 = N_3 \| N_e \| N_f$ where $N_3$ is specified as in the proof of Theorem 4, $N_e = N_\to \circ (N_=^{x_{1,2}, x_{3,2}} \| N_{=n}^{x_{1,3}})$ and $N_f = N_{\neq \frac{(n+1)((n+1)+1)}{2}+1}^{x_{1,2}}$ where $N_{\neq t}$ is analogous to the construction of $N_{=t}$ given in Lemma 5. Furthermore, we adjust $comb_4$ in layer $l_4$ to be represented by the FNN $relu(x_3 + \cdots + x_8 + x_9)$. We refer to the gadgets described in Lemma 4 and Lemma 5 as well as the proof of Theorem 1 for further details.

Consider the adjustment in $l_3$. FNN $N_e$ in $comb_3$ ensures that $T_{\mathcal{S},n}(w) = 1$ only if the row index corresponding to the last symbol is equal to $n$. Note that $N_3$ checks whether row and column index corresponding to the last symbol are equal. Additionally, $N_f$ checks if there is no id equal to $\frac{(n+1)((n+1)+1)}{2} + 1$. This corresponds to the position id of the successor of the vector representing tile $(n, n)$. Furthermore, the adjustment of $comb_4$ considers the output of $N_e$ and $N_f$ in addition to the outputs of $N_3$. In summary, we have that $T_{\mathcal{S},n}$ only outputs 1 given $w$ if the word length is such that the row index corresponding to the position of the last symbol of $w$ in a respective octant tiling is equal to $n$ (ensured by $N_e$), that $w$ is at most of length $\frac{(n+1)((n+1)+1)}{2}$ (ensured by $N_f$) and if $w$ represents a valid encoded tiling (the remaining parts of $T_{\mathcal{S},n}$).

Additionally, we need to argue that $T_{\mathcal{S},n}$ works as intended, despite the fact that it is limited by some FA $F$ using a representation size that is at most logarithmic in $n$. These arguments follow the exact same line as in the proof of Theorem 2, but using FA $F$ that uses $m = \lfloor 6\log(\max(|S|, n)) \rfloor + 2$ bits and handles overflow using saturation. The reason for the larger representation size is that words $w$ representing a valid encoded tiling ending at position $(n, n)$ are of length $|w| = \frac{(n+1)((n+1)+1)}{2} \leq n^2$. Thus, we use $\lfloor 4\log(n) \rfloor + 1$ integer bits to be able to represent a sum $\sum_{j=0}^i j = \frac{i(i+1)}{2} \leq i^2$

for all $i \leq n^2$ and $\lfloor 2\log(n)\rfloor + 1$ fractional bits to uniquely represent fraction $\frac{1}{l}$ for $l \leq n$. For detail see the proof of Theorem 2. Furthermore, the fact that we use $\lfloor 4\log(n)\rfloor + 1$ bits to encode integers and that $F$ handles overflow using saturation ensures that $N_f$ works as intended: we have that $\frac{(n+1)((n+1)+1)}{2} + 1 < n^4$ and, thus, we have that the id $\frac{(n+1)((n+1)+1)}{2} + 1$ occurs at most once, independent of the length of $w$ as it is not the point where $F$ enforces saturation on the positional embedding. Thus, $att_{\mathrm{self}}$ works for this position as intended and then $N_f$ checks the property described above correctly.

The argument that $T_{\mathcal{S},n}$ can be built in polynomial time is a straightforward implication from the arguments for Theorem 3 and the fact that $N_e$ and $N_f$ are a small gadgets with maximum parameter quadratic in $n$, which can be represented using a logarithmic amount of bits. $\qquad\square$

