# OpenReview forum: "Transformer Encoder Satisfiability: Complexity and Impact on Formal Reasoning"
_ICLR.cc/2025/Conference — ICLR 2025 Poster_

### Official Review · Reviewer_1jYG · 2024-10-28

**Soundness:** 3
**Presentation:** 3
**Contribution:** 2
**Rating:** 8
**Confidence:** 2

**Summary:**

The paper studies the complexity of deciding whether a transformer accepts a given input sequence. This problem of satisfiability (SAT) is studied on encoder-only transformers. Encoder-decoder transformers are not considered, as they are Turing complete and hence the SAT problem is immediately undecidable. Firstly, the SAT problem is proven to be undecidable for hardmax encoder-only transformers using a reduction to tiling problems, even when the transformer has log precision. Secondly, decidable restrictions are achieved using bounded length sequences or fixed precision in combination with periodic embedding results. However, in these cases, the SAT problem still remains NEXPTIME.

**Strengths:**

Although the content is very theoretical, the authors take care to first give a general overview of the results (Section 3), after which proof sketches are given (Sections 4 and 5) and more detailed proofs follow in the appendices. This presentation kept the paper relatively digestible, despite the fact I’m not very familiar with this subject matter. From a theoretic viewpoint, I found the results interesting. The practical usefulness seems somewhat limited to me, but as the authors note the presented SAT problem is foundational in relation to safety and verification of model properties and this work provides a useful start in its study.

**Weaknesses:**

As I am not really familiar with the computational complexity study of neural networks, I do not feel very confident to judge the significance of the findings. However, some of the assumptions seem to limit the practical usefulness of the results.
-  As the authors mention themselves, the distinctions between the considered transformer models fall away when a bounded word length is assumed. Given that transformers almost always have a (fixed) context window, such a bounded length assumption seems quite reasonable to me, and the results for unbounded length seem less relevant.
- The $\mathcal{T}_{udec}$ class for transformers (and its further restrictions) uses hard-attention instead of soft-attention. Previous works have indicated that soft-attention can achieve significantly different results compared to hard-attention (Strobl et al. 2024).
- The decidability results of e.g. Theorem 3 are based on naive enumeration, which is not realistic except for very short word lengths.

**Questions:**

- Do any of the proven results change if satisfiability is defined as exceeding a given threshold instead of just being equal to 1? This could be a more realistic condition for acceptance.

- Do the authors have an intuition on to what extent their results could generalize to soft-attention?

- It was not clear to me if the definition of SAT on transformers is new, or has been proposed before.

- This perhaps a bit of a broad/vague question, but seeing the many undecidability and hardness results proven in the paper, do the authors think that there is hope for formal verification techniques on transformers? Or should the verification efforts focus on more tractable and less expressive classes of models than the transformer?

*Some small nitpicks*

- Brackets for citations are not used properly. The paper consistently cites transformers Vaswani et al. (2018) instead of transformers (Vaswani et al., 2018).

- The font in Figure 2 is too small for me to be readable without zooming.

---

> ### Author Response · Authors · 2024-11-14
> **answer to questions**
>
> We appreciate you taking the time to review our paper! We are particularly grateful for your precise articulation of the contributions our paper makes, and for acknowledging our efforts to ensure its accessibility to a broader audience.
>
> Below, we provide succinct responses to each of your inquiries, and we look forward to engaging in a productive dialogue!
>
>
> ### Questions
> > Do any of the proven results change if satisfiability is defined as exceeding a given threshold instead of just being equal to 1? This could be a more realistic condition for acceptance.
>
> In summary, the answer is: no. In more detail:
> In our lower-bound proofs, we can similarly construct feedforward neural networks (FNN) to represent the output function in those proofs, such that they output a value of $\geq c$ for arbitrary (rational) thresholds $c$ if and only if the desired property is satisfied, instead of precisely equalling $1$.
> In the case of upper-bound proofs, the distinction is negligible, as determining whether a numerical value equals $1$ or is greater than or equal to some threshold $c$ involves a similar level of complexity.
>
> In addition, we agree with your suggestion to emphasise this point in our paper. Consequently, we have included a clarifying note as a footnote following the SAT definition. Thank you!
>
> > Do the authors have an intuition on to what extent their results could generalize to soft-attention?
>
> The upper bounds we established also apply to TE with softmax as normalisations. Thus, the interesting part here are lower bounds.
>
> We do not have a definitive answer for the lower bounds. It seems that the constructive approaches we employed to prove lower bounds might not work with softmax, due to its less discrete nature. For instance, we frequently leveraged specific encodings to ensure that hardmax could uniquely focus on particular positions. This is by definition impossible with softmax, as long as we assume unbounded precision. Consequently, we believe that alternative types of arguments would be necessary when considering softmax in lower bounds.
>
> > It was not clear to me if the definition of SAT on transformers is new, or has been proposed before.
>
> To the best of our knowledge, this problem has not been explored before in the context of transformers, at least in this specific manner. However, there is similar work for other models, such as Graph Neural Networks (GNN) and FNN; see references [1] and [2] for more details, but note that the terminology sometimes differs.
>
> > This perhaps a bit of a broad/vague question, but seeing the many undecidability and hardness results proven in the paper, do the authors think that there is hope for formal verification techniques on transformers? Or should the verification efforts focus on more tractable and less expressive classes of models than the transformer?
>
> We appreciate the question and agree with you that, from a practitioner's perspective, this is the primary concern within this area of research. At present, our results, especially the lower bounds, suggest that it is advisable to concentrate on less expressive models when aiming for sound and complete verification or interpretation. For those focused on transformers, the best approach right now may be to employ, for example, non-complete methods.
>
> However, this holds only for the time being. Our findings suggest that there are potential strategies to enhance the possibilities of formal reasoning, such as focusing on quantised models or significantly restricting the context length of interest.
> Additionally, a detailed examination of our lower bound proofs provides insights into the properties that may contribute to the generally high complexity of the problem. For example, using less powerful positional encodings or normalisation functions could yield more tractable settings.
>
> Nevertheless, the question remains whether such transformers will prove to be competitive and what the precise complexity of problems like SAT will be for these models. Ultimately, obtaining a better outcome than NP-completeness is improbable in non-trivial cases, as evidenced by similar challenges found in FNN [2]. It is plausible that the middle ground will consist of PSPACE-completeness, akin to certain non-trivial GNN models [1], which some consider a generalised form of transformers.
>
> > Some small nitpicks
>
> Thanks for pointing these out. We fixed both in the revised version of the paper.
>
> ---
> [1] Michael Benedikt, Chia-Hsuan Lu, Boris Motik, Tony Tan:
> Decidability of Graph Neural Networks via Logical Characterizations. ICALP 2024: 127:1-127:20
>
> [2] Marco Sälzer, Martin Lange:
> Reachability is NP-Complete Even for the Simplest Neural Networks. RP 2021: 149-164

---

> > ### Comment · Reviewer_1jYG · 2024-11-21
> >
> > Thank you for the clarifications! I currently have no further questions.

---

### Official Review · Reviewer_UiUG · 2024-10-30

**Soundness:** 3
**Presentation:** 2
**Contribution:** 3
**Rating:** 5
**Confidence:** 5

**Summary:**

This technical paper tackles the decidability of satisfiability for Transform encoders (TE), namely, does there exists an input such that the output of the TE is a given constant, say 1. This problem is a generic problem which can encode pattern recognition, etc. The results by the authors are as follows:

1) in general, the satisfiability problem is undecidable. The proof is by reduction to a tilling system, or equivalently to halting problem - a correct input of the TE leading to output 1 corresponds one to one to a (correct) halting unfolding of a run of the machine. The encoding requires a number of bits for each neuron computation logarithmic in the size of the input, and also that the embeddings are *not* periodic.

2) If the input is bounded by size n, then the problem is NP-complete(n) (leading to a usual NExptime-complete complexity if n is written in binary), the upper bound being trivial and the lower bound being the same reduction than for undecidability.

3) Alternatively, if the input is unbounded but the number of bits for each neuron computation is bounded by a number of bits part of the input, then the problem is also NEXPTIME-hard using the exact same encoding, although no upper bound, is provided in this case, not even decidability.

4) The most interesting decidability result is that if the number of bits for each neuron computation is bounded and the embedding periodic, with both the period and number of bits are part of the input (in unary), then the problem is decidable in NEXPTIME (but no lower bound is provided).

**Strengths:**

1. The paper explores the complexity of Transform encoders. Understanding the theoretical complexity of this hot topic is very timely.

2. The proofs seem solid. 2 constructions are non-trivial (one lower bound encoding (1) and one decidability & (upper bound) complexity proof (4), the other being direct application of the first construction).

**Weaknesses:**

1. The paper could be written in a more reader-friendly way, it is very technical. Statements of the theorems are unnecessarily complicated. The number of Theorems is also inflatted. E.g. theorem 2 should just be a note at the end of theorem 1: "undecidable in general, even when restricted to log-precision transformers." At the end of the day, there are 2 main results in this paper (1 and 4 listed above).

2. (edit: partially improved, but the solution does not fundamentally change the usability landscape, which is in practice somehow still restricted to bounded input) The biggest weakness is that the complexity landscape has a serious gap. The authors need 2 restrictions together to get decidability (see 4.), and the proof of undecidability needs both restriction lifted. So what happens if only one of this restriction? Even more problematic, the authors focus heavily on one of these restrictions (bounded-precision, which is arguably a very reasonable restriction to consider), leaving the other (periodicity, a much stronger restriction) as a technical factor that can be easily overlooked by an inattentive reader (for instance, the notation is using a small '_o'). The proof in appendix C reveals that its actually periodicity that is the main driver for decidability, and fix-precision only seems to be accessory to simplify the proof, and may be useful for the Nexptime upper bound complexity.

**Questions:**

1. Where is the fix-precision used in the proof of lemma 3? Is it necessary?

2. Do you have proof of decidability for fix-precision (*without* periodic embeddings)?

In all cases, you cannot draw Sat[T^fix] on the Nexptime ball (in figure 2), as there is no proof it is *in* Nexptime. (edit: fixed)

---

> ### Author Response · Authors · 2024-11-14
> **answer to questions + weaknesses - part 1**
>
> Thank you for dedicating your time to providing a review of our paper. We hope this serves as a sound foundation for a productive discussion.
>
> We shall first address your direct questions, following which we will comment on the weaknesses you have highlighted.
>
> ### Questions
>
> > Where is the fix-precision used in the proof of lemma 3? Is it necessary?
>
> First, a brief response:
> Refer to line 1067, where it's noted that $V$ represents the finite set of all numerical values expressible within fixed-width arithmetic.
> Afterwards, we use the bound $|V|$ in line 1087.
>
> Now, to provide an intuitive explanation:
> While the periodicity combined with the finite input alphabet limits the number of possible distinct vectors following the positional embedding, fixed-width arithmetic is essential for bounding intermediate computations of a TE. For instance, the hardmax or softmax normalisations used in the considered classes of TE exhibit averaging behaviour, meaning their outputs depend on the length of input words.
> This length is unbounded and, thus, there is a possibly infinite number of different outputs.
> However, since only finitely many values are representable (see set $V$), we observe an "internal periodicity" concerning normalisations based on the pigeon hole principle. The same intuition applies for pooling functions. In summary, this is the idea behind the bound used in line 1087 and why the fixed-width arithmetic assumption is essential in the proof of Lemma 3.
>
> > Do you have proof of decidability for fix-precision (without periodic embeddings)?
>
> Thank you for raising this question! No, we do not have a formal proof for this setting, but this is mostly due to the fact that fixed-width arithmetic generally leads to periodic embeddings:
>
> Let's consider positional embeddings that encode positional information in the form $1, 2, 3, \ldots$. Such embeddings are evidently periodic in fixed-width arithmetic, using wrap-around to handle overflow, as it results in $1,2, \dotsc, n, 1, 2, \dotsc, n, \dotsc$ and so on. If overflow is handled using saturation, these positional embeddings result $1, 2, 3, \ldots, n, n, \ldots$, comprising a finite prefix followed by a suffix with periodicity of $1$, for which the arguments are the same as we used.
>
> We acknowledge that alternative embeddings could be considered, which (presumably) do not witness periodic behaviour in fixed-width arithmetic. For instance, enlisting the digits of pi as positional information. However, we believe that such considerations are somewhat contrived and detract from the fundamental objectives of our paper.
>
> > In all cases, you cannot draw Sat[T^fix] on the Nexptime ball (in figure 2), as there is no proof it is in Nexptime.
>
> Thank you for pointing that out. This was not the intention we wished to communicate in the illustration. Originally, we aimed to clarify this in the figure's description with the statement: "The NEXPTIME-hardness result of $\text{Sat}[\mathcal{T}^\text{FIX}]$ is depicted by placing it precisely on the upper boundary between NEXPTIME and all decidable problems."
>
> Nevertheless, we recognise that this might still lead to some misunderstanding. Thus, we have updated the figure and the clarifying sentence in the description in the revised version which hopefully resolves any ambiguities.
>
> ### Weaknesses
>
> > The paper could be written in a more reader-friendly way, it is very technical.
>
> Thank you for considering the style of presentation we have adopted. This paper aims to enhance readability in three significant ways: Firstly, in Section 3, we provide an informal overview of our results and their implications, which should be accessible to those without specialised knowledge of the topic. Secondly, Sections 4 and 5 offer proof sketches for all our core results, allowing readers with more expertise to grasp the essence of our arguments. Lastly, the technical appendix contains the full detailed proofs, enabling the complete verification of all our findings.
>
> We believe this method effectively presents our specialised results to a wider audience. However, we remain open to any suggestions you might have!
>
> > Statements of the theorems are unnecessarily complicated.
>
> If you find the time, we would appreciate it if you can provide further clarification on this point of criticism, as it is not our intention to come across as overly complex. Most of our results are conveyed in a single sentence, such as "Problem X is undecidable." or "Problem Y is in NEXPTIME." While this does require a basic understanding of computational complexity theory, we believe it is the most succinct manner to present these findings.

---

> > ### Author Response · Authors · 2024-11-14
> > **answer to questions + weaknesses - part 2**
> >
> > > The number of Theorems is also inflatted. E.g. theorem 2 should just be a note at the end of theorem 1
> >
> > We acknowledge the connection between Theorems 1 and 2, as well as Theorems 4 and 5. However, we believe it is a matter of preference, and deem it reasonable to treat them separately. This is due to the distinct arguments involved, each of which is considerably detailed, as highlighted in the technical appendix.
> >
> >
> > > The authors need 2 restrictions together to get decidability (see 4.), and the proof of undecidability needs both restriction lifted. So what happens if only one of this restriction? ...
> >
> > This is addressed in our response to your second question. However, if you feel there are additional aspects that require clarification, we would be happy to elaborate on this further.

---

> ### Comment · Reviewer_UiUG · 2024-11-14
>
> Dear Authors,
>
> Thanks a lot for the answer.
>
> Some more details.
> > While the periodicity combined with the finite input alphabet limits the number of possible distinct vectors following the positional embedding, fixed-width arithmetic is essential for bounding intermediate computations of a TE.
>
>
> I believe you do not need to reason at the level of intermediate computations. Different use of Dirichlet lemma should provide decidability without finite alphabet (with possibly higher complexity than what you obtain with finite input alphabet).
>
>
> >we do not have a formal proof for this setting, but this is mostly due to the fact that fixed-width arithmetic generally leads to periodic embeddings
>
>
> I do not understand the logical fundation of such a sentence. The fact that periodicity is often used should not prevent you from making a proof without periodicity.
>
>
> Again, looking at things at a high level, a decidability proof seems reasonable with only fixed width arithmetic (but certainly with higher complexity than NEXPTIME-time).
>
>
> > we recognise that this might still lead to some misunderstanding
>
>
> You just cannot represent things this way (unless you have stronger results).
> This would be highly non-standard.
>
>
> > This is due to the distinct arguments involved
>
>
> Please, enlighten me to the argument involved in the proof of Theorem 2.
> As I see it, it is a simple: the encoding of the proof of Theorem 1 needs only logarithmic number bits, without (almost) any changes.
>
>
>
> > we would appreciate it if you can provide further clarification on this point of criticism,
>
> I was refering mainly to the statement of Theorem 3 (page 8), and to the explicit simplification of the statement of th1 and th2 into 1 single easier to parse statement.
>
> Statements of other Theorems are more reasonable, I rekon that I overgeneralized from these 3 theorems to all the theorems statements.
>
> For theorem 3 page 8: distinction un / bin is ok.
> Simplification could look like:
> BTRSATun[T ] is NP-complete.
> simple and effective.
>
> The additional "if Tudec ⊆ T then BTRSATun[T ] is NP-complete" brings absolutely no additional information AFAIK.
>
> So far, the authors did not bring strong additional information. The main concern remains about the big gap in the results, which I do not see much justification for. My current assessment remains, that is, interesting paper, but there is at least 1 missing result to make it ready for publication.

---

> > ### Author Response · Authors · 2024-11-14
> > **NP-completeness does not generically extend downwards**
> >
> > > For theorem 3 page 8: distinction un / bin is ok. Simplification could look like: BTRSATun[T ] is NP-complete. simple and effective.
> >
> > But unfortunately not correct for arbitrary $T$. The sentence would be true if it only stated "BTRSATun[T] is in NP". For NP-hardness to hold as well, $T$ needs to be "strong enough" to capture the inherent difficulty of NP-hard problems. There are many (trivial) classes $T$ of transformers for which BTRSATun[T] is *not* NP-complete, for instance transformers that realise constant functions.
> >
> > > The additional "if Tudec ⊆ T then BTRSATun[T ] is NP-complete" brings absolutely no additional information AFAIK.
> >
> > Yes, it does. The condition "$Tudec \subseteq T" is a syntactic (and therefore automatically checkable) property that guarantees NP-hardness.

---

> ### Author Response · Authors · 2024-11-18
> **addressing the gap between theorem 4 and 5**
>
> Firstly, we apologise for the delayed response to the remaining part of you comment.
> We have dedicated some time over the past days to discuss your query concerning the gap, and we believe we have a more satisfactory answer now.
>
>
> >I do not understand the logical fundation of such a sentence. The fact that periodicity is often used should not prevent you from making a proof without periodicity.
>
> We recognise that our initial comment might not have been clear enough. The intention behind our statement was:
>
> In a fixed-width arithmetic setting, every positional embedding must inherently witness periodicity.
> This is simply implied by the fact that the positional information used as input for the embedding must also be constrained by fixed-width arithmetic.
> Otherwise, the setting would be inconsistent.
>
> Therefore, the result of decidability that you inquired about is directly inferred by the arguments presented in Theorem 4, which we assumed was evident.
>
> However, we now recognise that this clarification was insufficient in the prior version of the paper.
> We have made the following changes:
>
> - We have enhanced Theorem 5 to explicitly include the result of decidability.
> - We revised the paragraph beginning at line 293 to better build an intuition for Theorem 5.
> - We incorporated arguments regarding decidability in the proof sketch of Theorem 5, beginning at line 501, and included these in the complete proof starting at line 1098 as well.
> - We adjusted Figure 1 accordingly.
>
> We believe this addresses the gap you mentioned earlier, and we thank you for your valuable insights!
>
> > You just cannot represent things this way (unless you have stronger results). This would be highly non-standard.
>
> We appreciate your suggestions here. Fortunately, this issue is resolved; the stronger version of Theorem 5 can now be included with clarity!

---

> ### Comment · Reviewer_UiUG · 2024-11-21
>
> > In a fixed-width arithmetic setting, every positional embedding must inherently witness periodicity. This is simply implied by the fact that the positional information used as input for the embedding must also be constrained by fixed-width arithmetic. Otherwise, the setting would be inconsistent.
>
> At the very least, the period itself is probably exponentially higher (or more) than if you force it.
> I will check the new proof.
>
> > We appreciate your suggestions here. Fortunately, this issue is resolved; the stronger version of Theorem 5 can now be included with clarity!
>
> This solves nothing. You CANNOT draw a problem in NEXPTIME if YOU HAVE NO PROOF IT IS IN NEXPTIME.
>
> You may (or may not) have close the decidability gap,
> but the complexity gap remains.

---

> > ### Author Response · Authors · 2024-11-21
> >
> > First and foremost, we want to reiterate our thanks for your willingness to thoroughly discuss the paper with us. We understand it requires a significant commitment of your time.
> >
> > > This solves nothing. You CANNOT draw a problem in NEXPTIME if YOU HAVE NO PROOF IT IS IN NEXPTIME.
> >
> > We apologise if there was a misunderstanding. Our intention was to convey that the new, revised version addresses the concern you had with the original depiction in Figure~1, not because we have now placed Theorem 5 within the NEXPTIME class, which would be false as you correctly stated. Rather, it is now placed within the decidable class.
> >
> > To avoid further misunderstanding, we have chosen not try to visually depict NEXPTIME-hardness anymore. We stated this in the description of the figure.

---

> ### Comment · Reviewer_UiUG · 2024-11-21
>
> >>Simplification could look like: BTRSATun[T ] is NP-complete. simple and effective.
>
> >>But unfortunately not correct for arbitrary. The sentence would be true if it only stated "BTRSATun[T] is in NP". For NP-hardness to hold as well, needs to be "strong enough" to capture the inherent difficulty of NP-hard problems. There are many (trivial) classes  of transformers for which BTRSATun[T] is not NP-complete, for instance transformers that realise constant functions.
>
> What I meant was BTRSATun[T ] is NP-complete for the set T of all transformers. (I admit the T I used was not clear).
> Which is 100% correct. It is actually implied by the current statement of Th1 as T_udec \subseteq T.
>
> It does not mean that all fixed Transformer would be hard to check - but neither does your proof or your statment show that. Hardness is ensured only for the family of T_S you handcrafted.
>
> Now, "BTRSATun[Tudec] is NP-complete" would be an additional slightly intereting information  only if Tudec were a strong and interesting restriction of the class T of all transformers. It does not worth the complicated statement. Hence as i wrote in the review: "unnecessarily complicated statement", and probably even misleading and hard to parse as non standard. The audience at ICLR only barely understand complexity, there is certainly no point making it appear more complex than it is for no good reason.
>
> Worst case, you should write a simple statement:
>
> " BTRSATun[T ] is NP-complete for the set T of all transformers. Additionnaly, the hardness holds even when the class of transformers is restricted to T_udec."
>
> Like that, you have the simple human understandable statement, plus a refinement for the happy few who cares (honestly, even while being in the field, I think the additional information is not necessary - if I need a stronger statment, I ll need to check the proof anyway to be sure it falls in the class I am considering).
>
> check e.g. Th4 of Jorge Pérez, Pablo Barceló, Javier Marinkovic: Attention is Turing-Complete. J. Mach. Learn. Res. 22: 75:1-75:35 (2021). It is written EXACTLY in this way. Because this is the standard way of stating such complexity results.

---

> > ### Author Response · Authors · 2024-11-21
> >
> > Let's summarise the discussion regarding the statement of Theorem 3 on page 8. Your final suggestion is
> > - BTRSATun[T] is NP-complete for the set T of all transformers. Additionnaly, the hardness holds even when the class of transformers is restricted to T_udec.
> >
> > In line 416-418 we write:
> >
> > - Let $\mathcal{T}$ be a class of TE. Then,
> >     - BTRSATun[$\mathcal{T}$] is decidable in NP and if $\mathcal{T}_\text{udec} \subseteq \mathcal{T}$ then BTRSATun[$\mathcal{T}$] is NP-complete.
> >     - [...]
> >
> > We genuinely appreciate your dedication to presenting results as clearly as possible. We share your commitment to this mission and we like your suggestion. However, do you agree, especially in this direct comparison, that it is mostly a matter of taste as to whether the upper or lower statement is more comprehensible? If you agree, we believe we have thoroughly explored this aspect of the discussion. If not, just say so and we happily respond in detail.
> >
> > ---
> >
> > Overall, we believe we have addressed all the concerns you have raised at this point, and we thank you once again for contributing to an engaging discussion and taking the rebuttal seriously. In particular, your insights have been valuable in clarifying the nuances surrounding Theorem 5 and strengthening the connection between Theorems 4 and 5. We feel this improved the paper.
> >
> > Would you let us know if you have any further questions?
> > If not, do you still feel that the revised version of the submission does not meet the threshold for publication?
> >
> > While our opinion is obviously biased, we feel that you overall appreciate our contributions and your main concerns are adressed by now.

---

> > ### Author Response · Authors · 2024-11-26
> >
> > > you should write a simple statement:
> > > " BTRSATun[T ] is NP-complete for the set T of all transformers. Additionnaly, the hardness holds even when the class of transformers is restricted to T_udec."
> >
> > So you are asking us to weaken Theorem 3 so that it only makes statements about *two* classes:
> > - NP-completeness for the class of all transformers.
> > - NP-hardness for the class T_udec.
> >
> > We would like Theorem 3 to continue to make a broader statement about *a multitude* of transformer classes, namely that NP-completeness holds for any class between T_udec and the class of all transformers (with respect to set inclusion). It seems like the standard way of doing this is something along the lines of
> >
> > "For any class of transformers T such that T_udec $\subseteq$ T we have that BTRSATun[T] is NP-complete."
> >
> > and it is not misleading, it is just stronger.

---

> ### Comment · Reviewer_UiUG · 2024-11-23
>
> >do you agree, especially in this direct comparison, that it is mostly a matter of taste as to whether the upper or lower statement is more comprehensible?
>
> I agree that both statments are equivalent.
> I do not agree that the one you have in the paper would be clearer for anyone (except may be for the authors).
> There is no reason to consider " $\mathcal{T}$ be any class of TE".
> This is non standard. The $\mathcal{T}$ be any class of TE" brings nothing to the statment.
> You only need 2: the most general one for which you have an algorithm with the stated complexity, and the most restricted one ensuring hardness.
>
>  Worse, this is misleading, as if the statment meant something else (which I thought it was).
>
> The standard formulation avoids the misleading nature, and is clear for everyone, because it meant exactly that and nothing more.
> Find me any paper *not from any of the authors* with such a circunvoluted formulation.
> Again, I opened the Perez et al paper, and the first theorem I looked at was written in the standard way, which I was kindly proposing (and many different formulation would be perfectly fine as well).
>
>
> Bottom line: I do not mind any formulation, as long as it is not misleading.
>
>
> Yours is.
>
>
> You are welcome to propose any non misleading one. It is important for a paper to be as clear and simple as possible, correctness of the statment is necessary but far from being sufficient.
>
>
> This was exactly the same case as drawing a class inside the  NEXPTIME circle while you had no proof that it was in NEXPTIME. This was misleading. Thankfully, you changed that.

---

> ### Comment · Reviewer_UiUG · 2024-11-27
>
> > you are asking us to weaken Theorem 3
>
> Let me make a simple analogy here.
>
> Imagine that you are proving that "a=b".
>
> You propose to state:
> Theorem: for all c, a+c = b+c.
>
> (which is correct by the way)
>
> I propose you to state:
> Theorem: a=b
> (or anything else simple and non misleading)
>
> You are claiming that this would weaken the Theorem...
>
> I am claiming that your statement is unecessarily complicated, and that the simple a=b is abslutely equivalent, and should anyone need 'for all c, a+c = b+c' (or anything else), it will be straigthforward from the simple statement.
>
> Further, the statement "for all c, a+c = b+c" is weird as this c is coming out of nowhere and has no specific purpose.
> This makes the reader wonder why, if there would be anything wrong with the obvious a=b statment. Henceforth, misleading.

---

> > ### Author Response · Authors · 2024-11-27
> >
> > We continue to appreciate your concern for raising clarity in our paper and your efforts to clarify your ideas. To respect your time, let's try to streamline the discussion a little.
> >
> > __Our standpoint on this matter__
> >
> > The contributions our paper aims to make involve demonstrating specific complexity results for different classes of TE. While Theorems 1 and 2 address undecidability for certain classes, Theorems 4 and 5 establish decidability along with both upper and lower complexity bounds for others. Theorem 3 serves an intermediate role:
> >
> > By putting aside the general SAT problem and focusing on bounded-SAT, the specific class of TE is irrelevant, as long as general assumptions such as efficient computability are guaranteed, for NP-membership. And, hardness is provided once a certain level of expressiveness is assured.
> >
> > Your suggestion weakens the role Theorem 3 plays in the overall story of the paper, as it references solely the class of all TE, not all classes of TE.
> >
> >
> > __Direct questions to you__
> >
> > - Is the style of Theorem 3 on page 8 the primary factor causing you to think our paper is not ready for acceptance?
> > - If so, do you think we could find a middle ground where Theorem 3 continues to include that NP membership is given for all classes? We would suggest:
> >     - "For all classes of TE $\mathcal{T}$ problem $\text{btrSATun}[\mathcal{T}]$ is in NP and as soon as $\mathcal{T}_\text{udec} \subseteq \mathcal{T}$ it is NP-hard."

---

> ### Comment · Reviewer_UiUG · 2024-11-27
>
> > Is the style of Theorem 3 on page 8 the primary factor causing you to think our paper is not ready for acceptance?
>
> I would not say your paper is not ready for acceptance.
>
> The changes you have made made it ready for acceptance in a good conference, just not in a top conference such as ICLR.
> I'd say the results are interesting but not meaningful enough for ICLR, they are a bit below the bar.
>
> You did not fundamentally change the contributions of your paper.
> On one hand, you syntactically added a decidability result, but on the other hand, it does not change fundamentally what you are proposing. Concretly, you have:
>
> - undecadibility for the general class of TE, which is not surprising and does not bring fundamentally different methodology than the reduction in Perez 2021, although the result is certainly new. The same construction implies all the complexity lower bounds in your papers. Interesting but not suprising.
>
> - The undecidability relies on unbounded counters, which is not reasonable in practice (as Perez et al. This is a good thing actually).
>
> - On the decidability resuts: All the decidability results would in practice only be meaningful for bounded inputs. Indeed, Transformers with periodic encodings would only be accurate for period = O(size of input), otherwise you have too many collision and the position of symbols is lost (which means that the technically harder decidability result on periodic TE is probably of limited interets).
>
> - The class of fixed arithmetic TE is immediatly a periodic one (with a large period), making it not a meaninfgul new result.
>
> - In practice, inputs are bounded. The complexity for this class is important, but the proof is straigthforward.
> The problem remains that the NP complexity is in terms of the size of the input. This implies untractable complexity for any input that is not small.
>
> It remains two technically complicated proofs (th1 and th4). Interesting, deserves publication in a good but not top conference.
>
> That being said, this is my take on it, and I will not oppose acceptance if other reviewers are deeply convinced and interested in the results.
>
> > Your suggestion weakens the role Theorem 3 plays in the overall story of the paper, as it references solely the class of all TE, not all classes of TE.
>
> Theorem 1,2,4,5 are written in the standard way, looking at ONE (as generic as possible) class of TE.
> Theorem 3 departs from that for no good reason.

---

> > ### Author Response · Authors · 2024-11-27
> > **thank you for the discussion!**
> >
> > While we (obviously) not fully agree with all of your evaluations, we feel that our discussion has reached a natural conclusion.
> >
> > Thus, it is only left to say thanks for the considerable time you have dedicated to discussing the paper with us.

---

### Official Review · Reviewer_RHMx · 2024-11-01

**Soundness:** 2
**Presentation:** 1
**Contribution:** 2
**Rating:** 3
**Confidence:** 4

**Summary:**

The paper studies the decidability of the emptiness problem of the language recognized by a transformer encoder classifier,  that is, whether there exists a string w in the input domain for which the given transformer will return true. It also studies the complexity of such problem in contexts where it is found to be decidable.

**Strengths:**

The main interest of the paper is to try to define bounds on the complexity of decidable classes of problems related to languages recognized by transformes.

**Weaknesses:**

The paper itself is quite a high-level description of a list of results. The necessary definitions, such as the tiling problem, and other key aspects of the proofs are in the appendix or completely missing. The absence of important definitions makes the paper difficult to follow.

Besides, the structure of the paper with a preview of the results does not enhance readability but quite the contrary. It would be much readable if the results were shown, proved and explained once, providing reasonably detailed proofs that put forward the important issues (while the much technical details are put in the appendix).

**Questions:**

Pg. 2. L 54-54. I disagree with this comment. Programming languages are Turing Complete, still formal reasoning and verification has been a very active, productive and necessary field of computer science. Nevertheless, it is worth knowing	the boundaries of decidability and complexity to figure out methods to cope with that.  Please comment.

Pg. 2. L 66-69. What do you mean by “formal interpretation”? The term is not common in the field of formal verification and you do not define it here nor provide references to definitions or related work. An example is provided later in Pg. 4, Sec. 3, which reduces to a verification problem. There is missing related work regarding formal methods and tools for extracting automata-based models from different kinds of neural classifiers (including transformers) and language models that should be cited here. Such methods can accomplish verification and generate explanations. Good sources of references for this matter are the Proceedings of the International Conference on Grammatical Inference 2023 and the last two LearnAut workshops. That line of relevant work is not referenced by the cited papers.

Pg. 3. L 169-174. What part of the TC proof of Pérez et al made you believe that languages recognized by the class T_udec could possibly be decidable?

Pg. 4. Could it be possible that softmax rather than hardmax change the decidability result?

Pg. 5. L 251. The 3. Clearly, bounded satisfiability is decidable if T(w) is computable because you can enumerate all words up to length n. The important result here is the complexity bound. Also, the way this theorem is written  here is different from Sec. 5. You should rewrite this.

Pg. 5. L. 265. In what sense fixed-width arithmetic has a similar effect to bounding the input length? Besides, why not considering then T^FIX only?

Pg. 6. L. 302. Why is it the case that T_udec is the weakest class in terms of expressivity?

Pg. 6. L. 306. The name “octant” tiling problem does not seem to be standard. It is not mentioned as is in the provided reference. Also it is not clear why it is necessary to distinguish between “tiling” problem and “tiling word” problem. The definitions in the appendix do not make this clear. Could you explain it?

Pg. 8. L. 401. It reads “one can reasonably assume the size of a syntactic representation of T to be polynomial on |T|”. It is not clear to me that this assumption is reasonable. Please provide arguments.

Pg. 8. L. 402. The so-called polynomial evaluation property is not discussed in Section 3 nor elsewhere in the paper. Why is it reasonable?

Pg. 8. It seems that the proof sketch of The 3 does not take into account the representation of w. Please comment on this.

Other comments

Pg. 3. L. 138. k should be L since i is a layer, and layers go from 1 to L.

Pg. 3. L. 157. different rational number”s”

---

> ### Author Response · Authors · 2024-11-13
> **answer to questions - part 1**
>
> Firstly, we would like to express our gratitude for your comprehensive review! We appreciate the considerable time you devoted to raising those questions.
>
> We are looking forward to an engaging discussion. To make it efficient, we have paraphrased the main questions below.
> However, the numerous questions you stated make it necessary to spread our answer across several comments.
>
> ---
>
> ### Questions
>
> > Pg. 2. L 54-54.  "I disagree with this comment. Programming languages are Turing Complete, still [...]. Please comment."
>
> We agree. It is certainly worth delving into research on formal reasoning for encoder-decoder transformers. The comment to which you refer was intended to
> clarify our decision to focus on encoder-only models, as this is the first work of its kind. We aim to establish fundamental results without imposing
> unnecessary restrictions. Commencing with full encoder-decoder capabilities could potentially require artificial constraints on models and settings to
> achieve decidability, due to their already known high expressive power (Peréz et al. 2021). Thus, we selected encoder-only models for this study, but never
> intended to suggest that encoder-decoder transformers lack interest.
>
> >Pg. 2. L 66-69. "What do you mean by “formal interpretation”?"
>
> The term "formal" indicates our focus on methods that are both sound and complete. We elaborated on this in the paragraph beginning on line 38. "Formal
> interpretation" refers to problems, procedures, and related topics concerning interpretability properties of neural network-based models. For a reference,
> please see Marques-Silva & Ignatiev (2022), where the authors employ the term "formal XAI". We discussed this in the paragraph beginning on line 190.
>
> > Pg. 3. L 169-174. "What part of the TC proof of Peréz et al. [..] T_udec could possibly be decidable?
>
> We view the principal contribution of this work as establishing a foundational framework for comprehending the computational complexity of formal reasoning
> within transformers. The class T_udec under consideration should be regarded as the so far tightest "threshold" concerning expressive power, above which
> formal reasoning of transformer encoders is impossible/undecidable.
> In the study by Peréz et al. (2021), the authors concentrated on encoder-decoder transformers, whereas our research is dedicated solely to encoder-only
> transformers. While some of our architectural choices for T_udec were inspired by their work, primarily to establish the aforementioned threshold, the
> models we investigate are of a different type.
>
> > Pg. 4 "Could it be possible that softmax rather than hardmax change the decidability result?"
>
> We understand this question as you referring to the class $T_\circ^{\text{fix}}$ or the result of Theorem 4. The class
> $T_\circ^{\text{fix}}$ encompasses transformer encoders (TE) that can incorporate either softmax or hardmax normalisations. Consequently, the
> upper bound on decidability also applies to classes of TE analogous to $T_\circ^{\text{fix}}$ that exclusively employ hardmax, which means that
> the presence of softmax is not the reason.
>
> > Pg. 5. L 251. The 3. "Clearly, bounded satisfiability is decidable [...] The important result here is the complexity bound."
>
> We agree with your suggestion. This informal restatement of Theorem 3 would indeed benefit from the inclusion of the upper bounds specified in the formal
> theorem. We have addressed this in the revised version by stating NEXPTIME- and NP-completeness.
>
> > Pg. 5. L. 265. "In what sense fixed-width arithmetic has a similar effect to bounding the input length? Besides, why not considering then T^FIX only?"
>
> The sentence you are referring to was meant to convey that both, bounding the input length or considering fixed-width arithmetic, lead to decidability with
> an NEXPTIME upper bound.
> The relevance of the bounded input length setting lies in its practicality as a natural constraint, such as when context sizes are limited.
>
> > Pg. 6. L. 302. Why is it the case that T_udec is the weakest class in terms of expressivity?
>
> While much remains unknown, providing a comprehensive hierarchy of classes for transformer encoders is challenging. This difficulty arises from the
> numerous parameters involved, such as variations in encoding, normalisations, and scorings, among others.
> The statement you are referring to was meant to introduce T_udec as the class for which we currently know, based on the results of our paper, that it is
> the weakest leading to the undecidability of SAT. However, you are correct: the way that that sentence is currently phrased could suggest a stronger claim
> than intended. We have addressed this in the revised version.

---

> ### Author Response · Authors · 2024-11-13
> **answer to questions - part 2**
>
> > Pg. 6. L. 306. The name “octant” tiling problem does not seem to be standard. It is not mentioned as is in the provided reference. Also it is not clear
> why it is necessary to distinguish between “tiling” problem" and “tiling word” problem. The definitions in the appendix do not make this clear. Could you
> explain it?
>
> Octant tiling is a variant of the well-known quadrant-tiling (named in [Rob71]) instance in the class of domino tiling problems, where the tiling is
> restricted to a certain octant of the plane, also known as a quadrangle. It is true that there does not seem to be one citable reference for
> the undecidability of the octant tiling problem, but this is also due to the fact that it is easily seen to be undecidable by observing that the
> standard reduction from the halting problem for Turing Machines to the quadrant tiling problem yields an octant tiling (because after n steps a TM can
> have modified or visited at most the first n tape cells). The octant tiling problem is often used in the literature for undecidability proofs, for
> instance in [BGD+10,BDG+14,B22,B24,HMS08].
>
> In our undecidability proofs, the "octant tiling word problem" serves as an intermediate problem between octant tiling and the satisfiability problem for
> transformer encoders. The "octant word tiling problem" is used to translate the source problem in the reduction, formulated as a constraint problem
> over a two-dimensional geometry, into a constraint problem over a one-dimensional geometry, i.e. as a problem of finding a linear word structure. This is
> all there is to it.
>
> > Pg. 8. L. 401. It reads “one can reasonably assume the size of a syntactic representation of T to be polynomial on |T|”. It is not clear to me that this
> assumption is reasonable. Please provide arguments.
>
> To ensure that Theorems 3 and 4 are still broadly applicable, we have defined classes of transformer encoders (TE) without rigidly fixing every
> single parameter such as the precise combinations of scorings and similar aspects. However, for obtaining complexity bounds it is essential to establish the
> measure of representation size for transformers. The size measure |T| used in the paper takes only the most significant parts of an encoder transformer into
> account, namely alphabet size, depth, width, and dimensionality. This ensures that Theorems 3 and 4 are applicable to transformer models that may vary in
> certain aspects, for as long as the other parameters do not exceed this polynomial bound. It is, of course, possible to devise transformers that do not meet
> the assumption in question, for example by use of non-standard normalisation functions that require large programs or very large constants for their
> computation or representation. To the best of our knowledge, the transformers that are widely considered in the literature and in practice do all meet this
> property of being representable in space that is polynomial in the measure |T|. This is why we call this assumption reasonable.
>
> > Pg. 8. L. 402. The so-called polynomial evaluation property is not discussed in Section 3 nor elsewhere in the paper. Why is it reasonable?
>
> Thank you for pointing this out; it appears to be an oversight on our part. The polynomial evaluation property is deemed reasonable as it aligns with how
> encoders process their output. The outlined procedure is as follows: for each layer $l_i \leq L$, the output sequence $w^i = x_1^i \dotsb x_{|w|}^i$ is
> derived from $w^{i-1}$ by pairing each $x_h^{i-1}$ and $x_k^{i-1}$ with $h, k \leq |w|$. This pairing occurs within each attention head
> $\mathit{att}_{i,j}$ where $j \leq H$. Given that $|T|$ is dependent on both $L$ and $H$, this computation is polynomial in terms of $|T| + |w|$.
> We added a clarifying paragraph after Theorem 2 in Section 3!
>
> ---
> [Rob71] R. Robinson. "Undecidability and Nonperiodicity for Tilings of the Plane", Inventiones Mathematicae, 12(3), 1971 pp. 177–209
>
> [B22] L. Bartholdi, "Monadic second-order logic and the domino problem on self-similar graphs", Groups Geom. Dyn. 16 (2022), pages 1423–1459
>
> [B24] B. Bednarczyk, "Exploring Non-Regular Extensions of Propositional Dynamic Logic with Description-Logics Features", Log. Methods Comput. Sci. 20(2) (2024)
>
> [BDG+10] D. Bresolin, D. Della Monica, V. Goranko, A. Montanari, G. Sciavicco, "Undecidability of the Logic of Overlap Relation over Discrete Linear
> Orderings", Electronic Notes in Theoretical Computer Science, vol. 262, 2010, pages 65-81
>
> [BDG+14] D. Bresolin, D. Della Monica, V. Goranko, A. Montanari, G. Sciavicco, "The dark side of interval temporal logic: marking the undecidability
> border", Ann. Math. Artif. Intell. 71(1-3): 41-83 (2014)
>
> [HMS08] I. M. Hodkinson, A. Montanari, G. Sciavicco, "Non-finite Axiomatizability and Undecidability of Interval Temporal Logics with C, D, and T", CSL
> 2008:308-322

---

> ### Author Response · Authors · 2024-11-13
> **answer to questions  - part 3 + weaknesses pointed out**
>
> > Pg. 8. It seems that the proof sketch of The 3 does not take into account the representation of w. Please comment on this.
>
> We believe you are referring to the fact that $|w|$ measures only the length of a word, as opposed to its representation size, which should also account
> for $|\Sigma|$. You are indeed correct in this observation. However, the size of a TE $|T|$ does depend on $|\Sigma|$, hence this poses no issue for the
> lower bounds demonstrated in Theorem 3.
> Nevertheless, we agree with your point and have included a clarifying sentence in the revised version in the first paragraph of the proof sketch of Theorem
> 3 (page 8).
>
> > Other comments
>
> Thank you for pointing out these typos. We fixed them in the revised version!
>
>
> ### Weaknesses
>
> Regarding the structure and presentation of the paper: we see the point. But other reviewers have commented on the presentation in an opposite way. We
> will write a short reply on this issue to all reviews as a general comment.

---

> ### Author Response · Authors · 2024-11-23
>
> Dear Reviewer RHMx,
>
> We want to reiterate our gratitude for your numerous questions from the initial review. We believe we have addressed them adequately thus far, leading to improvements in clarity! Thank you!
>
> As we near the conclusion of the rebuttal phase, please let us know if there is anything specific you would like to discuss. Since your main concern seems to be the overall style of presentation, we would be happy to discuss this with you in greater detail.
> However, as the end of the rebuttal phase approaches, we may not have enough time to address this thoroughly if we do not begin soon.

---

### Official Review · Reviewer_1Qh5 · 2024-11-05

**Soundness:** 3
**Presentation:** 3
**Contribution:** 2
**Rating:** 6
**Confidence:** 5

**Summary:**

The paper proves several hardness results on the satisfiability of encoder-only Transformers. These results demonstrates the hardness of preforming formal verification of satisfiability over Transformers Encoders, which is undecidable for unbounded-length, log-precision Transformer Encoders, NEXPTIME-Hard for bounded-length inputs and bounded precision.

**Strengths:**

Overall, the theoretical contributions of this work can be impactful as formalizations of impossibility results of general formal verification over Transformers. The theoretical contribution of implementing a tiling system within Transformer Encoders can also be useful for further theoretical work, whereas prior works on expressiveness on Transformer Encoders mostly focused on upper bounds with circuit complexity. As such, I recommend for Acceptance (assuming the authors make appropriate clarifications as mentioned below).

**Weaknesses:**

The paper tries to make arguments that connects the satisfiability theorems proven in the paper to “formal reasoning” (i.e, model verification and interpretation), which I believe is not sufficiently justified. In the section 3.1 “Satisfiability as a baseline formal reasoning problem” the author makes 2 examples: robustness verification and formal interpretation. However, both examples require the input to satisfy certain properties and decides satisfiability on the set of inputs with the given properties. It is unclear whether the how hardness results still hold when input space is constrained as in the given examples. As such, this connection between the theorems proven in the paper and “formal reasoning” should be accurately characterized.

The naming in the paper can cause much confusion. SAT typically refers to the Boolean Satisfiability problem in computational complexity, and using SAT to also refer to Transformer Encoder Satisfiability can be confusing to many readers, especially the claim that “SAT is undecidable” in the abstract. It is recommended to use a different acronym for the specific problem. Similarly, “Formal Reasoning” also has specific meanings referring to reasoning over formal systems with well-defined inference rules and axioms. The “formal reasoning” in this paper can be directly stated as “verification and interpretation”.

At the current state, the paper’s conclusions are not fundamentally surprising given the recent line on work on the expressiveness of Transformers (although, as mentioned, formalizing such statements and providing a concrete construction is a sufficient contribution). From a practicality perspective, both NP-Hard and NEXPTIME-hard are both infeasible, and it would be more impactful if the work shows interesting classes of Transformers/properties that can be verified in polynomial time.

**Questions:**

Please address the concerns raised in the Weaknesses section.

---

> ### Author Response · Authors · 2024-11-14
> **comments on weaknesses - part 1**
>
> We are grateful for your time in reviewing our paper.
> In response to the weaknesses you highlighted, we will address each one step by step, aiming for a productive and engaging discussion.
>
> ### Weaknesses
>
> > The naming in the paper can cause much confusion. SAT typically refers to the Boolean Satisfiability problem in computational complexity, and using SAT to also refer to Transformer Encoder Satisfiability can be confusing to many readers, especially the claim that “SAT is undecidable” in the abstract.
>
> We understand the potential for confusion in terminology. To address this, we have consistently renamed the problem as TrSAT throughout the paper in the revised version to distinguish it clearly from the Boolean satisfiability problem. Additionally, we have rephrased the introductory sentence in the appendix. We appreciate your suggestion!
>
> > The paper tries to make arguments that connects the satisfiability theorems proven in the paper to “formal reasoning” (i.e, model verification and interpretation), which I believe is not sufficiently justified. [...]
>
> As you rightly noted, we have devoted Section 3.1 (line 177-) to highlight this connection stating examples regarding robustness of interpretation problems. There we provide two examples of a connection. In the following following we give a third, more general one:
>
> A prominent problem in the category of formal verification is the reachability problem (Reach), which can be described as follows: given a transformer $T$, a set of valid inputs $X$, and a set of valid outputs $Y$, decide whether there exists an input $w \in X$ such that $T(w) \in Y$. The TrSAT problem can be considered to be a fundamental form of Reach, where the set of valid inputs $X = \Sigma^*$, with $\Sigma$ representing the alphabet of $T$, and the set of valid outputs is $Y = \{1\}$.
> The basic nature of TrSAT implies that Theorems 1 and 2 apply to all variants of Reach, where we can define $X$ and $Y$ as mentioned above. We believe this represents a level of expressiveness in input and output specifications which is general. Similarly, upper bounds such as those in Theorems 4 and 5 are valid for all variants of Reach, provided that the complexity of determining membership in $X$ or $Y$ does not overshadow the complexity introduced by the kinds of transformers considered.
>
> Nonetheless, you highlight an important aspect: research centred on specific reasoning problems must elucidate the connections to SAT.
> We believe this task is straightforward for those engaged in such areas due to the inherent simplicity of SAT. The two examples in the paper
> and the one given above highlight this. Apart from the general concepts already discussed both above and in our paper, we see limited value in defining further, more specific connections in this work.
> Ultimately, the aim of this paper is to lay down foundational baselines, which we believe should be framed as broadly as possible.
>
> >  “Formal Reasoning” also has specific meanings referring to reasoning over formal systems with well-defined inference rules and axioms. The “formal reasoning” in this paper can be directly stated as “verification and interpretation”.
>
> We have introduced the phrase "formal reasoning" to have a single term to collectively refer to problems, procedures, and similar concepts involved in "formal verification and formal interpretation." We explained this in lines 44–45. That is all there is to it, but we remain open for other suggestions.
>
> Nonetheless, to avoid any misunderstandings we have included an additional sentence in the beginning of the appendix of the revised version of the paper to clarify the term as early as possible.

---

> > ### Author Response · Authors · 2024-11-14
> > **comments on weaknesses - part 2**
> >
> > > The paper’s conclusions are not fundamentally surprising given the recent line on work on the expressiveness of Transformers.
> >
> > While we understand your perspective, we respectfully offer a deviation in our opinion. Although, as indicated by results like [1], one might anticipate undecidable cases (Theorem 1), our research further extends these findings even into log-precision settings (Theorem 2). Additionally, our analysis of the effects of quantisation and periodic embeddings (Theorems 4 and 5) presents insights that are not immediately derived from the current advancements in comprehending the expressive power of transformer encoders.
> >
> >
> > > It would be more impactful if the work shows interesting classes of Transformers/properties that can be verified in polynomial time.
> >
> > We concur that demonstrating such results would indeed be profoundly impactful. However, considering the substantial expressive capabilities of transformer encoders, it is improbable that such results exist without either trivialising the class of transformers or the class of properties being addressed. For instance, even for feedforward neural networks common verfication problems are NP-hard [2].
> > As a result, these expectations may be somewhat ambitious.
> >
> > Nonetheless, we concur that pursuing more tractable scenarios is desirable. In this regard, we perceive our work as a foundational endeavour, laying a groundwork for future research aimed at identifying tractable contexts for formal reasoning. Most of all: our work presents principal limitations on the possibility to find tractable classes of transformers/properties.
> >
> >
> > ---
> >
> > [1] Jorge Pérez, Pablo Barceló, Javier Marinkovic:
> > Attention is Turing-Complete. J. Mach. Learn. Res. 22: 75:1-75:35 (2021)
> >
> > [2] Marco Sälzer, Martin Lange: Reachability is NP-Complete Even for the Simplest Neural Networks. RP 2021: 149-164

---

> > > ### Comment · Reviewer_UiUG · 2024-11-21
> > > **undecidability and logarithmic encodings**
> > >
> > > Sorry to hijack Reviewer  1Qh5 thread, but I had the same question, and it s probably better to ask it here.
> > >
> > > > our research further extends these findings even into log-precision settings (Theorem 2).
> > >
> > > After checking [1] Jorge Pérez, Pablo Barceló, Javier Marinkovic: Attention is Turing-Complete. J. Mach. Learn. Res. 22: 75:1-75:35 (2021), the proof techniques seems extremely close to the one you employ.
> > > In particular, all [1] needs is a counter different for each configuration, with number of configurations = O(size of the world).
> > > Encoding the counter in logarithmic size is quite straightforward.
> > > I agree that there is no such statment in [1] though.
> > >
> > > Could you point us to a particularity of the proof of [1] which would make the undecidability holding under log precision hard to make?
> > >
> > > I agree that the decidability class are new in your paper though.
> > > But the undecidability is a large part of your paper.

---

> > > > ### Author Response · Authors · 2024-11-21
> > > >
> > > > No worries, we appreciate your strong participation in the rebuttal.
> > > >
> > > >
> > > > > After checking [1] (...) the proof techniques seems extremely close to the one you employ.
> > > >
> > > > We respectfully want to state that we see this statement as an oversimplification:
> > > >
> > > > First, we should not forget the key distinction between our work and [1]: in [1], Turing-completeness is demonstrated for a class of sequence-to-sequence transformers, lets call it $\mathcal{S}_{[1]}$, or encoder-decoder transformers, whereas our focus is solely on encoder transformers. Consequently, the model we examine is at most as expressive as the one in [1]. Note that this statement is a bit informal as we compare sequence-to-sequence with sequence-to-vector.
> > > > However, it implies that Theorem 1 is no direct implication of [1], and a new proof is necessary.
> > > >
> > > > Second, one can see, for instance, in Figure 1 of [1], that the authors' arguments in [1] strongly depend on the decoding capabilities of encoder-decoder transformers of $\mathcal{S}_{[1]}$. These capabilities simply do not exist in the classes we consider, as we only use encoders. Consequently, the constructions utilised there cannot be applied to our case.
> > > >
> > > > The familiarity you may perceive arises from the fact that we intentionally designed our class, $\mathcal{T}_\text{udec}$ used in Theorem 1 (and later restricted in Theorem 2), drawing inspiration from the encoder components used in [1]. Thus, at a basic level, there are similarities like the way the expressive capabilities of positional embeddings are exploited, and so forth.
> > > > However, we want to emphasise that we clearly stated in the submission our intentional choice of this class to get upper complexity bounds for trSAT.
> > > >
> > > > > In particular, all [1] needs is a counter different for each configuration, with number of configurations = O(size of the world). Encoding the counter in logarithmic size is quite straightforward. I agree that there is no such statment in [1] though.
> > > >
> > > > We are unsure which specific part of the construction you are referring to. Especially since the setting of [1] and ours does differ in more than just "a single counter". Especially, the decoding capabilities exploited in [1] do not exist in our work.
> > > > Nevertheless, feel free to share the exact arguments you have in mind.
> > > >
> > > > > Could you point us to a particularity of the proof of [1] which would make the undecidability holding under log precision hard to make?
> > > >
> > > > The primary particularity lies in the models we discussed earlier. Specifically, [1] utilises encoder-decoder transformers, whereas our paper exclusively focuses solely on encoders. This foundational difference makes addressing the question as you likely intended quite difficult, without considering Theorem 1 of our paper as an intermediate step.
> > > >
> > > > The step from Theorem 1 to Theorem 2 involves demonstrating that the construction detailed in Theorem 1 operates as intended, even within the constraints of the log-precision setting. Achieving this necessitates an analysis of the construction.
> > > > This is precisely what is undertaken in the proof of Theorem 2. But we already discussed this.
> > > >
> > > > We partly agree with your earlier implications that, given Theorem 1, the proof of Theorem 2 is in large part busy work. But, as we want to state our results thoroughly, it is necessary work. Besides, it definitely does no harm to include it. Furthermore, Theorem 2 serves to complement the recent findings of Merrill et al. concerning the expressiveness of log-precision encoders, which makes it valuable in the range of results presented in our submission, especially for those parts of the community interested in such settings.

---

> > > > > ### Comment · Reviewer_UiUG · 2024-11-21
> > > > >
> > > > > > the authors' arguments in [1] strongly depend on the decoding capabilities of encoder-decoder transformers of $\mathcal{S}_{[1]}$.
> > > > >
> > > > > Thanks, that is a convincing answer. I think you should emphasis that (more?) in your paper.

---

> > > > > > ### Author Response · Authors · 2024-11-21
> > > > > >
> > > > > > We are glad to have resolved this concern!
> > > > > >
> > > > > > We added a clarifying sentence in the related work section, where we discussed the work of Peréz at al. (see line 79) .
> > > > > > Additionally, we underlined the encoder-only nature of the model we consider in the introduction (see line 54).

---

> ### Author Response · Authors · 2024-11-27
>
> Dear reviewer 1Qh5,
>
> In your original review, you stated:
>
> > As such, I recommend for Acceptance (assuming the authors make appropriate clarifications as mentioned below).
>
> We have since addressed the concerns you highlighted and would like to inquire if any issues remain regarding your initial review.
> Should there be any remaining concerns, we would happily discuss them further, especially given the extended period now available for discussion.
>
> In particular, your original score appears to be a "weak accept", which seems somewhat at odds with the statement you made above.
> It is possible that your score reflects an inclination to first ensure that the highlighted weaknesses are thoroughly addressed.
>
> Are there any remaining concerns or reasons preventing you from considering our submission as deserving of a "clear accept"?
> We are keen to address any remaining issues you may see with our submission.

---

### Author Response · Authors · 2024-11-14
**comment on assessment of our style of presentation**

The initial reviews presented contradicting perspectives on the style of presentation, some describing it as too technical, others as not technical
enough. We understand each of these viewpoints. They may arise naturally given that the paper's topic lies at the intersection of machine learning and
theoretical computer science, respectively computational complexity. To address the broader audience present at ICLR, presumably mostly at home in the machine learning area, we decided to present our results in three stages:

- *Overview*: Initially, we present an overview of the core results, accompanied by informal explanations of their implications. This part should be
accessible to the entire audience.
- *Proof sketches and ideas*: Next, we offer for all core results a description of the proof strategies, including the underlying ideas and tools, as well as sketches of the arguments. This is designed to help readers gain an initial understanding of the proofs and an insight into their validity.
- *Technical appendix*: Finally, we provide complete technical proofs for all statements, constrained by a strict page limit, with major sections placed in the appendix. Naturally, this requires some background knowledge regarding the topics of the paper.

The overview is Section 3, sketches are Sections 4 and 5 and technical parts are in the appendix.

---

### Comment · Reviewer_UiUG · 2024-11-23
**New Theorem 5 decidability proof seems incorrect.**

Edit: it depends on some definition fine print details which are not clear (see more in my next reply below).

I was just checking the new decidability claim for fix width (but without periodicity).

> a fixed-width setting with b bits enforces a periodicity of size at most 2^b

This seems incorrect to me.
Consider the following b bits positional encoding;

position => bits 4b to 5b of position.

ex with b=3.

000

000

000

....

000

001

001

...

001

010

010

...

010

011

...


This would be periodic, but the period would be >2^{4b}.

,
Isnt it a counter example of the claim?


Worse, you can use "bits of position from X to X+b" (above, we have X=4b) for any X you want.

---

> ### Author Response · Authors · 2024-11-23
>
> Thanks for checking the revised version of the paper!
>
> ---
>
> You make a mistake here regarding the overall setting:
>
> Assume some TE $T$ is given, using an embedding $\mathit{emb}(a_i,i)$
> where $a_i$ is a symbol and $i$ the position of symbol $a_i$ in some word $w = a_1 a_2 \dotsb a_m$
>
> Now, assume that $T$ computes $T(w)$ in a fixed-width arithmetic $F$ using $b$ bits to represent numerical values. We denote the representation of some number $i$ in $F$ by $[i]_F$.
>
> Then, the first thing $T$ does in the computation of $T(w)$ is to compute
> $\mathit{emb}(a_1, [1]_F),
> \mathit{emb}(a_2, [2]_F), \dots$ and so on. Namely, $T$ embeds $w$. But what happens when $\mathit{emb}(a_i, [i]_F)$ for some $i > 2^b$ should be computed? Obviously,  $i$ cannot be represented exactly in $F$. Thus, an overflow occurs.
> As the sequence of positions $i$ is increasing $1, 2, \ldots$ wrap-around or saturation both lead to periodicity.
>
> We hope this clears things up for you.

---

> ### Comment · Reviewer_UiUG · 2024-11-24
>
> I perfectly understand what you mean. Let me make the matter clear: It s a matter of definition, and it is very important: Is the (symbol, position) part of the fixed arithmetic, or does the fix arithmetic only starts from the embedding, after writing the (symbol, position) fully?
>
> The informal description you wrote in line 62 seems to relate more to the latter :
>
> >a quantized TE, meaning TE whose parameters and internal computations are limited by some fixed-width arithmetic
>
> I thought this was a very good idea. Transformers are nowadays computed internally using around 8bit arithmetic for efficiency reasons. Crucially, this is doable because quantizing internally embeddings and internal computations only loose very slight accuracy. Even if the TE is not quantized, you could verify its quantized version, which would be already a reasonably strong trustworthy statement. Last, 8 bits is not too big, 2^8=256, so that even a NEXPTIME complexity (n=8) = NP(n=256) would seem plausibly verifiable with powerful tools. Hence the very good idea.
>
> Now, if you are also placing the symbol i of input within this fixed arithmetic bound, looking at 8 bits makes no sense. This would destroy accuracy of the Transformer, as the position of symbol would be totally lost for any large enough input, with too many symbols with the same quantized position. A reasonable bound could be 64 in that case (64bit architecture). But then, we are looking at 2^64 >10^18. Now, n=64 is certainly unachievable for a NEXPTIME algorithm (= NP(n=10^18), even EXPTIME(n=64) is not reasonable). Intuitively, this brings little more than the result on bounded input – which your Theorem 3 already states.
>
> For all these reasons,  I had in mind the definition where input position is NOT quantized, for which your decidability proof does not hold (the counterexample I provided).
>
> I agree that for the definition you have in mind (input position IS quantized), the decidability proof holds – trivially I may add from invoking Theorem 4 directly. But also it is not interesting because of complexity reasons, while the quantized AFTER (symbol, position) would be much more meaningful (that is what I expected, and it is certainly harder to prove).
>
> Some further questions:
>
> Did you double check that the lower bound part of Theorem 5 holds with this definition of Fixed arithmetic (including the input symbols and positions?
>
> Can’t you come up with an upper complexity bound rather than just decidability?

---

> > ### Author Response · Authors · 2024-11-24
> >
> > Thank you again for taking extensive interest in our work! To value your time we try to
> > streamline the discussion and answer concisely.
> >
> > ---
> >
> > > [...] while the quantized AFTER (symbol, position) would be much more meaningful [...]
> >
> > Thank you for suggesting an alternative perspective regarding transformer encoders
> > operating within a practically realised arithmetic.
> > We concur that the approach you described could be quite interesting,
> > especially in a study focused on quantised settings.
> > However, the context of our work is considerably broader, namely gaining a general understanding
> > of the complexity of SAT regarding TE in various forms.
> >
> > We have chosen a fundamental setting, making the sole assumption that practically realised
> > transformers are constrained by some form of arithmetic, be it fixed- or floating-point,
> > employing $b$ bits for numerical representation. To ensure consistency within this setting,
> > everything must be constrained, including the positional values.
> >
> >
> > > [...] and it is certainly harder to prove.
> >
> > We would not expect it to need fundamentally different arguments.
> > It is more a matter of clearly defining which class of embedding functions
> > we consider and how they are practically realised.
> >
> > Two examples:
> >
> > In the case of all strictly monotonic embedding functions, you can employ similar
> > reasoning as in our context, specifically utilising the pigeonhole principle
> > combined with overflow management. In such instances, the method of practical realisation does not significantly impact the outcome.
> >
> > For (non-strict) monotonic functions, it is necessary to examine their
> > implementation more closely. For instance, consider your example, which
> > is monotonic and repeats each value $k$ times (if we did not misunderstand you).
> > It is reasonable to assume that $k$ is stored as a numerical value. Than, $k$ is part of the EOT and
> > therefore its representation size. Then, you can use the same arguments as we used in Theorem 4.
> >
> > > Did you double check that the lower bound part of Theorem 5 holds with this definition of Fixed arithmetic (including the input symbols and positions)?
> >
> > To avoid any misunderstanding, the definition you are referring to has always aligned with our interpretation.
> > Therefore, from our perspective nothing changed for the lower bound of Theorem 5.
> >
> > Nevertheless, we are happy to answer your question. Informal and straight to the point, the reason is as follows:
> >
> > NEXPTIME-hardness comes from the bounded octant word tiling problem for binary $n$.
> > Thus, the respective TE only needs to accept words up to length $n^2$ and reject as soon as $n^2+1$ occurs. For details why this is the case,
> > we refer to the proof of Theorem 5. We used at least $6\log(n)$ bits in the arithmetic, which is more than enough
> > to represent positions $1, 2, \dotsc, n^2, n^2+1$.
> >
> > > Can’t you come up with an upper complexity bound rather than just decidability?
> >
> > As periodicity $p$ is, in the worst-case scenario, constrained by $2^b$,
> > where $b$ represents the number of bits, you get with the arguments of Theorem 4
> > an upper bound in NEEXPTIME (nondeterministic double-exponential time).
> >
> > But, to be frank, we do not really see the relevancy of this bound in the
> > overall story of our paper.
> >
> > To obtain more precise bounds, we would need to conduct more in-depth investigations,
> > as exemplified in the two examples above. Notably, this would heavily blow up
> > Theorem 5, as we would need to differentiate based on the actual form of
> > the embedding functions.
> >
> > The arguments we chose are more basic.
> > Furthermore, we believe the examples above make clear, that it is not much of a challenge to come to a
> > conclusion for specific embeddings due to the general arguments and tools we provide. Thus,
> > those interested in these detailed complexity bounds should be able to build up on our investigation without much difficulty.

---

> ### Comment · Reviewer_UiUG · 2024-11-25
>
> 1) you have to edit the informal description you wrote in line 62 seems to relate more to the latter :
>
> >a quantized TE, meaning TE whose parameters and internal computations are limited by some fixed-width arithmetic
>
> Your argument and result hold on fixed arithmetic architecture, not quantized TE. As I explained, this is much less interesting.
> And fixed arithmetic is indeed what you use in the definition.
> It is important to be very clear in the paper to not mislead readers.
>
> 2)
> > We have chosen a fundamental setting,
>
> I dont see why fixed arithmetic would be more fundamental than quantized TE. Both seems equaly fundamental to me.
> Actually, fixed arithmetic is more restricted than quantized TE, as any TE on fixed arithmetic would be naturally quantized by the number of bits of the fixed arithmetic. This does not look more fundamental to me.
>
>
> 3) in the version I had in mind (and which is somehow intuitive with quantized TE), the function from (input, position) to embeding would NOT BE limited by the quantization, and your argument would not apply. To be clear, you would have a full embedding to quantized embedding, and start computing on the quantized embedding. That is what happens with TE which use 8bit arithmetics.
> Again, you do not want to produce the full embeding using 8 bit arithmetic, as you would loose most of the positional information.
>
>
> 4) nondeterministic double-exponential time: I do not understand why you wouldn't want to write that.

---

> > ### Author Response · Authors · 2024-11-25
> >
> > Thanks for getting back to us in such timely manner.
> >
> > ---
> >
> > > you have to edit the informal description you wrote in line 62
> > > a quantized TE, meaning TE whose parameters and internal computations are limited by some fixed-width arithmetic
> >
> > Thank you for the suggestion. We have rephrased it to "... quantized TE, meaning TE
> > whose parameters and computations are carried out in a fixed-width arithmetic,". This should
> > remove the false focus on "internal" computations.
> >
> > > I dont see why fixed arithmetic would be more fundamental than quantized TE. Both seems equaly fundamental to me
> >
> > We understand that the setting you describe is of interest in practice, and agree that in a work focusing on quantised
> > settings it should be considered. But in this paper it would make things unnecessarily complicated.
> >
> > Here is an informal reason for this:
> >
> > Earlier, you mentioned that quantising the positional information could 'destroy the accuracy' of a transformer.
> > Let's consider the setting you propose, where we use $b$ bits for every computation the TE $T$ performs, but the
> > positional information $i$ used as input for the embedding can be arbitrarily large. To have a well-defined
> > framework, we assume that with the first basic arithmetic operation, such as $i \cdot c$ or $\frac{i}{2}$,
> > which most likely occurs during the embedding process of $T$, the result is represented using $b$ bits. This means,
> > a potentially large $i$ (beyond $2^b$) is immediately 'quantised'.
> >
> > Our research focuses on establishing foundational complexity bounds,
> > which necessitates understanding the foundational expressive capabilities of
> > specific classes of TE. Based on the explanation provided earlier, we assert
> > that allowing the positional index $i$ to be arbitrarily large does not fundamentally
> > impact these capabilities.
> >
> > However, you are right that when considering topics like the efficiency of inference in LLMs,
> > the difference between 64-bit and 8-bit precision is significant.
> > In a study focused on more fine-grained complexity results pertaining
> > to different variants of quantisations in TE, this should definitely be examined.
> >
> > > nondeterministic double-exponential time: I do not understand why you wouldn't want to write that.
> >
> > We think that it unnecessarily obscures the original purpose of Theorem 5, namely showing that there are
> > cases for which satisfiability is NEXPTIME-hard. Especially, since it can be immediately derived from
> > the decidability argument.

---

> > > ### Comment · Reviewer_UiUG · 2024-11-27
> > >
> > > > which most likely occurs during the embedding process
> > >
> > > Again, the embedding process would use full precision. Quantization would only occur *after* the embedding process (that's what happens AFAIK in practice). Even if it is not, as the operations are different for different scalar in the embeding vector, the proof would not apply.
> > >
> > > Anyway, I understand you are not tackling quantized TE in the paper, and this should be clearer in the paper (which is important), so let us stop the discussion about that off-topic.
> > >
> > > > We think that it unnecessarily obscures the original purpose of Theorem 5, namely showing that there are cases for which satisfiability is NEXPTIME-hard.
> > >
> > > You are perfectly free to weaken your statement, and I agree that this is not the main contribution.
> > > I will also be perfectly free to comment that the complexity is not stated in the Theorem.

---

### Author Response · Authors · 2024-12-04
**summary of rebuttal phase (authors perspective)**

We think it will help reproducibility of the rebuttal phase if we briefly summarize the individual threads and give final comments on it.

---

### reviewer RHMx (score 3, no rebuttal participation)

Regrettably, reviewer RHMx was not actively involved in the rebuttal phase. This was unexpected and somewhat disappointing, given that in the initial review, RHMx posed numerous questions concerning specific details of our work, which we have addressed in great detail starting [here](https://openreview.net/forum?id=VVO3ApdMUE&noteId=y7zbuJeHiu).

Considering this, the weaknesses highlighted by RHMx exclusively pertain to the style of presentation we employed. We have provided a comprehensive response concerning this issue [here](https://openreview.net/forum?id=VVO3ApdMUE&noteId=44QSypELkX) and are convinced that our style of presentation is suited to present our results to a broader community, while allowing interested experts to grasp all technical detail.

---

### reviewer UiUG (score 5)

We are grateful to reviewer UiUG for the extensive discussion and the substantial time investment made.

UiUG kindly summarised the rationale behind their final assessment of our submission as
falling slightly below the threshold [here](https://openreview.net/forum?id=VVO3ApdMUE&noteId=EEDoN0MCpP), which can be succinctly described as "captivating contributions, though lacking in practical relevance".

This conclusion somewhat perplexed us, as the focus on "practical relevancy" was
not apparent in either the initial [review](https://openreview.net/forum?id=VVO3ApdMUE&noteId=tLs8ZMWDr7)
or in our thorough discussions (refer [here](https://openreview.net/forum?id=VVO3ApdMUE&noteId=Crw4tMfP2d), [here](https://openreview.net/forum?id=VVO3ApdMUE&noteId=W9KUg1Hggw) or [here](https://openreview.net/forum?id=VVO3ApdMUE&noteId=GdM4ff8Wgp)), which mostly revolved around intricacies of our complexity results.

Nonetheless, UiUG explicitly stated in their summarised conclusion:

> That being said, this is my take on it, and I will not oppose acceptance if other reviewers are deeply convinced and interested in the results.

---

### reviewer 1Qh5 (score 6, no rebuttal participation)

Regrettably, reviewer 1Qh5 also did not participate actively in the rebuttal phase.

This is particularly unfortunate as we addressed their questions in considerable depth [here](https://openreview.net/forum?id=VVO3ApdMUE&noteId=ORRrEKephH), and they noted in their initial review that

> As such, I recommend for Acceptance (assuming the authors make appropriate clarifications as mentioned below).

which suggests that reviewer 1Qh5 was generally persuaded by our work, but remained at a 'weak accept', anticipating further clarification on certain issues to advance to a definitive accept.

---

### reviewer 1jYG (score 8)

We thank reviewer 1jYG for their active participation in the rebuttal process and are happy to have addressed all initially raised questions, as detailed [here](https://openreview.net/forum?id=VVO3ApdMUE&noteId=RQS93hde0T).

We are particularly happy to see that reviewer 1jYG highlights that our complexity focus on SAT as a baseline problem is

> foundational in relation to safety and verification of model properties and this work provides a useful start in its study

For details, see the original [review](https://openreview.net/forum?id=VVO3ApdMUE&noteId=bbkK5lYvhO).

---

### Meta-Review · Area_Chair_8oP3 · 2024-12-19

**Metareview:**

This paper provides complexity results for transformers, in particular, the authors identify the TRSAT problem and show its general undecidability. Practical cases are identified that can actually be decided.

The reviewers agree that the topic is important and timely and that the paper provides a solid, albeit not too strong, contribution to the ICLR community. Moreover, the authors are encouraged to reorganize and improve the paper according to the suggestions.

**Additional Comments On Reviewer Discussion:**

The discussion was extensive, with interaction with a part of the reviewers. The points that were raised by the reviewers were sufficiently addressed by the authors.

---

### Decision · Program_Chairs · 2025-01-22

Accept (Poster)